# RNA localization and co-translational interactions control RAB13 GTPase function and cell migration

Konstadinos Moissoglu[1,§], Michael Stueland[1,§], Alexander N Gasparski[1,§] (iD), Tianhong Wang[1,§],

Lisa M Jenkins[2,§], Michelle L Hastings[3] (iD) & Stavroula Mili[1,*,§] (iD)

## Abstract

**Numerous RNAs exhibit specific distribution patterns in mammalian cells. However, the functional and mechanistic consequences are relatively unknown. Here, we investigate the functional role of RNA localization at cellular protrusions of migrating mesenchymal cells, using as a model the RAB13 RNA, which encodes a GTPase important for vesicle-mediated membrane trafficking. While RAB13 RNA is enriched at peripheral protrusions, the expressed protein is concentrated perinuclearly. By specifically preventing RAB13 RNA localization, we show that peripheral RAB13 translation is not important for the overall distribution of the RAB13 protein or its ability to associate with membranes, but is required for full activation of the GTPase and for efficient cell migration. RAB13 translation leads to a co-translational association of nascent RAB13 with the exchange factor RABIF. Our results indicate that RAB13-RABIF association at the periphery is required for directing RAB13 GTPase activity to promote cell migration. Thus, translation of RAB13 in specific subcellular environments imparts the protein with distinct properties and highlights a means of controlling protein function through local RNA translation.**

**Keywords** antisense oligo; co-translational interaction; RAB13; RABIF; RNA localization

**Subject Categories** Membranes & Trafficking; Translation & Protein Quality

**The EMBO Journal (2020) 39: e104958**

## Introduction

Localization of RNAs to diverse subcellular destinations is widely observed in various cell types and organisms (Meignin & Davis, 2010; Medioni *et al*, 2012; Buxbaum *et al*, 2015). However, in mammalian cells, the functional and mechanistic consequences are relatively unknown.

In some cases, RNA accumulation can be accompanied by a corresponding increase in protein concentration at the same location. Such local protein gradients can be reinforced through translationally silencing RNAs prior to arrival at their destination (Besse & Ephrussi, 2008), thus ensuring tight spatial and temporal control of protein production and preventing deleterious effects of premature or ectopic translation (Jung *et al*, 2014; Buxbaum *et al*, 2015). This type of regulation has been described in highly polarized cells, such as neurons. For example, translational activation of RNAs localized at growth cones and the consequent increase in local protein abundance underlie axonal pathfinding decisions (Leung *et al*, 2006b; Colak *et al*, 2013; Wong *et al*, 2017). Similarly, activation of dendritic synapses upregulates translation of localized transcripts and is important for synaptic plasticity (Yoon *et al*, 2016; Rangaraju *et al*, 2017; Holt *et al*, 2019). Indeed, RNA localization appears to direct enrichment in neurites of almost half of the neurite-enriched proteome (Zappulo *et al*, 2017). A similar significant correlation between steady-state RNA and protein localization has been described in epithelial cells for proteins associated with organelles, such as mitochondria and the endoplasmic reticulum (Fazal *et al*, 2019).

Nevertheless, a concordance between RNA localization and protein distribution is not always observed. One case in point concerns RNAs enriched at dynamic protrusions of mesenchymal-migrating cells. RNA localization at protrusions is important for protrusion stability and cell migration (Mili *et al*, 2008; Mardakheh *et al*, 2015; Wang *et al*, 2017). However, there is little correlation between RNA and protein distributions (Mardakheh *et al*, 2015) and protrusion-enriched RNAs can be similarly translated in both internal and peripheral locations (Moissoglu *et al*, 2019), thus raising the question of what the functional role of RNA transport in these cases is. Here, we investigate the consequences of local peripheral translation focusing on the *RAB13* RNA.

RAB13 is a member of the Rab family of small GTPases which play important roles in vesicle-mediated membrane trafficking (Ioannou & McPherson, 2016; Pfeffer, 2017). It is amplified in the majority of cancers, and its levels inversely correlate with prognosis (Ioannou & McPherson, 2016). Activation of RAB13 at the

1   Laboratory of Cellular and Molecular Biology, Center for Cancer Research, National Cancer Institute, NIH, Bethesda, MD, USA
2   Laboratory of Cell Biology, Center for Cancer Research, National Cancer Institute, NIH, Bethesda, MD, USA
3   Center for Genetic Diseases, Chicago Medical School, Rosalind Franklin University of Science and Medicine, North Chicago, IL, USA
    *Corresponding author. Tel: +1 240 760 6844; E-mail: voula.mili@nih.gov
    §This article has been contributed to by US Government employees and their work is in the public domain in the USA

plasma membrane is required for cell migration and invasion (Ioannou *et al*, 2015), potentially through multiple mechanisms, including activity-dependent recycling of integrins or modulation of actin-binding proteins at the leading edge (Sakane *et al*, 2012, 2013; Sahgal *et al*, 2019). *RAB13* RNA is prominently localized at protrusive regions of multiple cell types (Mili *et al*, 2008; Feltrin *et al*, 2012; Moissoglu *et al*, 2019) together with a group of RNAs whose localization is regulated by the adenomatous polyposis coli (APC) protein and detyrosinated microtubules (Wang *et al*, 2017). We have previously examined the translational regulation of the *RAB13* RNA, showing that it is similarly translated in both internal and peripheral locations. Interestingly, translation of the *RAB13* RNA at the periphery is dynamically regulated with the RNA being actively translated at extending protrusions, while undergoing silencing at retracting regions. Thus, peripheral RAB13 translation appears to be functionally linked with protrusive activity (Moissoglu *et al*, 2019).

We show here that *RAB13* RNA and protein distributions are quite discordant, with *RAB13* RNA being enriched in the periphery, while RAB13 protein assumes mostly a perinuclear distribution. To assess the functional role of peripheral RNA localization, we devise a way to specifically prevent localization of *RAB13* RNA at peripheral protrusions without affecting its translation, stability, or the localization of other co-regulated RNAs. Importantly, we show that peripheral RAB13 translation does not affect the overall distribution of the protein or its ability to associate with membranes but is required for activation of the GTPase and for efficient cell migration. Our data show that RAB13 associates co-translationally with the exchange factor RABIF. Peripheral translation is required for RABIF-RAB13 interaction at the periphery and for directing RAB13 GTPase activity to promote cell migration. Our results indicate that translation of RAB13 in specific subcellular environments imparts the protein with distinct properties, thus highlighting a means of controlling protein function through local RNA translation.

# Results

## RAB13 RNA and protein exhibit distinct subcellular distributions

In both mouse and human mesenchymal cells, *RAB13* RNA is prominently enriched at peripheral protrusions (Fig 1A, and Mili *et al*, 2008; Wang *et al*, 2017). Our prior work has shown that peripheral *RAB13* RNA is actively translated at extending protrusions and silenced at retracting tails (Moissoglu *et al*, 2019). To assess whether translation of peripheral *RAB13* RNA leads to a corresponding increase in RAB13 protein, we visualized the distribution of endogenous RAB13. Interestingly, despite the peripheral *RAB13* RNA enrichment, at steady state, RAB13 protein is prominently concentrated around the nucleus (Fig 1B and C). However, since these cells are randomly migrating, some peripheral regions are in the process of retracting, thus likely containing silent *RAB13* RNA (Moissoglu *et al*, 2019). To enrich for actively extending protrusions, we grew cells on microporous filters, induced them briefly to migrate toward the bottom surface, and assessed protein and RNA distributions between fractionated protrusions (Ps) and cell bodies (CB) (Fig 1D). Consistent with the imaging data above,

*RAB13* RNA is significantly enriched at extending protrusions while, still, RAB13 protein is not (Fig 1D). We additionally considered whether acute stimulation might lead to a transient increase in peripheral RAB13 protein, since RNA translation can be locally induced upon activation of specific cell surface receptors (Huttelmaier *et al*, 2005; Cagnetta *et al*, 2018; Koppers *et al*, 2019). Again, however, we do not observe any increase in the amount of RAB13 protein at protrusions or in the overall RAB13 protein levels, upon stimulation with serum (Fig EV1). Moreover, the reported half-life of RAB13 protein is several hours (Schwanhausser *et al*, 2011; Boisvert *et al*, 2012; Mathieson *et al*, 2018). It exists in the cytosol, in a pool that is expected to diffuse rapidly, and also associates with intracellular vesicles playing a role in vesicle trafficking. Therefore, we think it is reasonable to assume that most of the lifetime of a RAB13 protein would be spent away from its site of synthesis. Overall, while we cannot exclude the presence of an undetectable pool of peripheral protein with distinct regulation, we think that these results strongly suggest that at least a significant proportion of the protein translated from peripheral *RAB13* RNA does not persist at the periphery but assumes a steady-state perinuclear distribution.

## A GA-rich motif within the mouse Rab13 3′UTR is necessary for localization at protrusions

To understand the functional role of peripheral translation, we first sought to narrow down on specific localization sequences. We had previously shown that a 200–300-nt region of the mouse *Rab13* 3′UTR is sufficient for localization and can competitively inhibit the localization of other peripheral, APC-dependent RNAs (Wang *et al*, 2017), suggesting that it contains a binding site for a factor commonly bound to APC-dependent RNAs. Using sequence alignment and gazing, we noticed a particular GA-rich motif, with the consensus RGAAGRR (where R is a purine), which is present, in one or multiple copies, in the 3′UTR of the majority (~ 60%) of APC-dependent RNAs (Figs 2A and EV2) and which is significantly enriched [$P = 4.99e\text{-}5$; motif enrichment analysis (meme-suite.org)] in APC-dependent RNAs compared to APC-independent RNAs, an RNA group which is also enriched at protrusions but through a distinct pathway (Wang *et al*, 2017). To test for any functional significance, we expressed an exogenous RNA carrying either the wild-type Rab13 3′UTR or the 3′UTR carrying specific deletions of this motif (Fig 2A and B). We imaged RNAs using single-molecule FISH and measured a Peripheral Distribution Index (PDI) to quantify their distributions in multiple cells (Stueland *et al*, 2019). Consistent with previous observations (Wang *et al*, 2017), a control β-globin RNA shows a mostly diffuse cytoplasmic distribution (Fig 2B and C), while addition of the Rab13 3′UTR is sufficient to promote its peripheral localization, denoted by low and high PDI values, respectively. Interestingly, deletion of one RGAAGRR motif (Rab13 UTR (Δ1)) significantly perturbed the ability of the Rab13 3′UTR to direct localization of the β-globin RNA, while deletion of two of them (Rab13 UTR (Δ1 + 2)) had a stronger effect making the distribution of the reporter more similar to that of the non-localized control (Fig 2C). An endogenous localized RNA (*Ddr2*) remained similarly localized at protrusions in all conditions (Fig 2C). Therefore, at least some of the RGAAGRR motifs are required for peripheral localization.

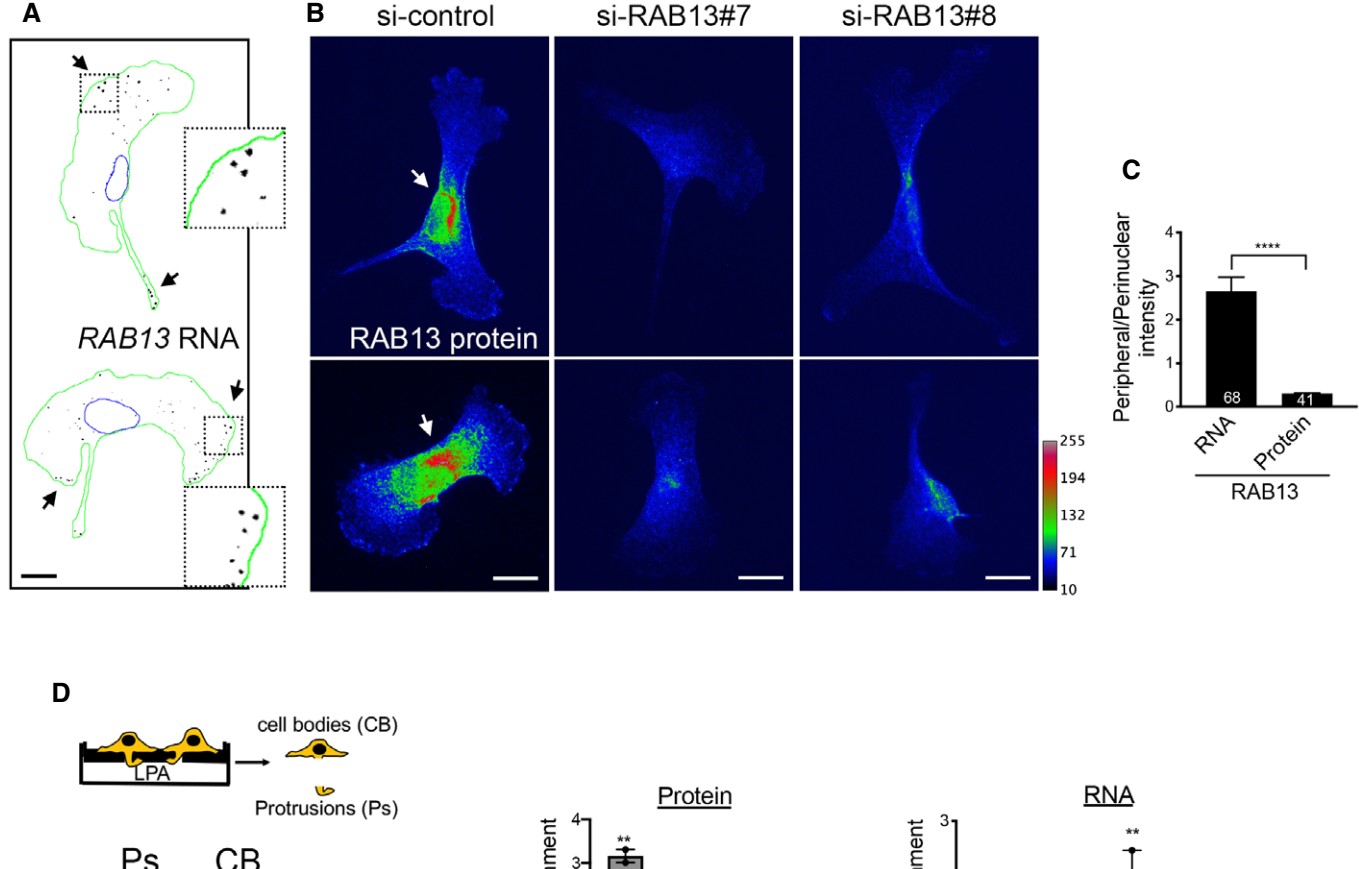

**Figure 1. RAB13 RNA and protein exhibit distinct subcellular distributions.**

A   Representative FISH images showing *RAB13* RNA distribution in MDA-MB-231 cells. Nuclei and cell outlines are shown in blue and green, respectively. Arrows point to *RAB13* RNA concentrated at protrusive regions. Boxed regions are magnified in the insets.

B   Representative immunofluorescence images of RAB13 protein in cells transfected with the indicated siRNAs. Reduction of intensity in RAB13 knockdown cells confirms the specificity of the signal. Arrows point to perinuclear RAB13 protein. Calibration bar shows intensity values.

C   Ratios of peripheral/perinuclear intensity calculated from images as shown in (A) and (B). Bars: mean ± s.e.m. Values within each bar represent number of cells observed in 3 independent experiments.

D   Protrusions (Ps) and cell bodies (CB) of cells induced to migrate toward LPA were isolated and analyzed to detect the indicated proteins (by Western blot; left panels) or RNAs (by RT-ddPCR; right panel). Ps/CB enrichment ratios from 2 independent experiments are shown. Bars: mean ± s.e.m. The enrichment of pY397-FAK serves to verify the enrichment of protrusions containing newly formed adhesions in the Ps fraction.

Data information: *P*-values: **< 0.01; ****< 0.0001 by Student's *t*-test (C) or analysis of variance with Dunnett's multiple comparisons test against GFP (D). Scale bars: 10 μm.

## Antisense oligonucleotides against the GA-rich region specifically interfere with localization of Rab13 RNA

Another notable feature of these motifs is that the majority of them (62%; 148 of 239 motifs in 3′UTRs of mouse APC-dependent RNAs) are found within more extended GA-rich regions, which exhibit high GA content (> 75%) for at least 30 consecutive nucleotides or more (Fig EV2). To further investigate the roles of these different features

and to, at the same time, interfere with the localization of the endogenous *Rab13* RNA, we used antisense oligonucleotides (ASOs), which can interfere with RNA structure formation or RNA–protein binding (Hua *et al*, 2010; Lentz *et al*, 2013; Havens & Hastings, 2016) (Fig 3A). Here, we utilized 25-nt-long phosphorodiamidate morpholino (PMO) ASOs.

We first delivered fluorescently labeled PMOs to determine the efficiency of delivery and their persistence in cells. PMOs were taken

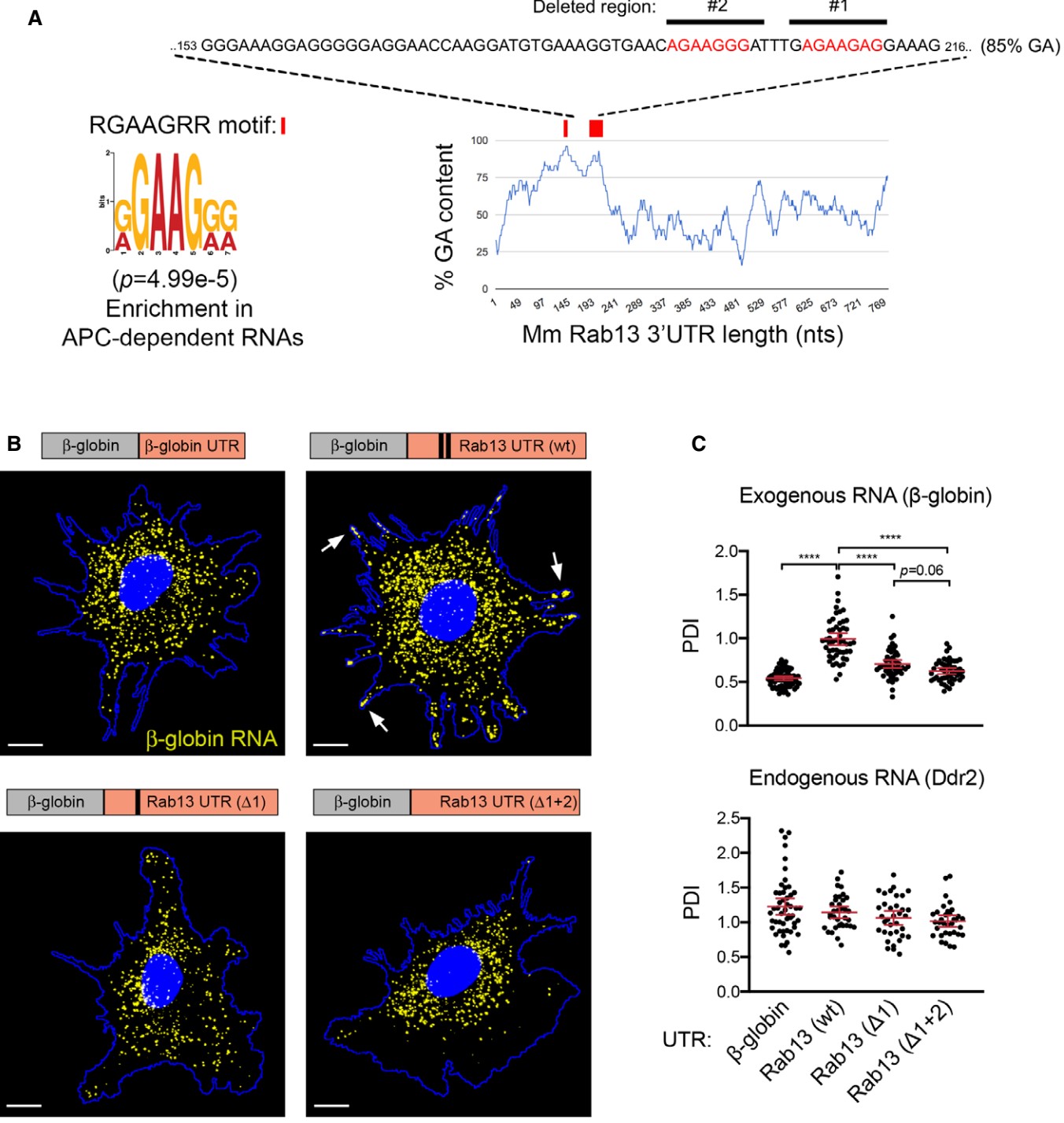

**Figure 2. A GA-rich motif in the mouse *Rab13* 3′UTR is necessary for localization at protrusions.**

A   Schematic showing the %GA content along the mouse *Rab13* 3′UTR using a 30-nt window size. Occurrences of the consensus GA-rich motif are indicated by a red rectangle. The exact sequence between nucleotides 153–216 is shown with GA motifs in red and deleted regions indicated by black bars. *P*-value by Fisher's exact test with Bonferroni's correction.

B   FISH images of mouse fibroblasts expressing the β-globin coding sequence followed by the indicated UTRs. β-globin RNA is shown in yellow. Nuclei and cell outlines are shown in blue. Arrows point to β-globin RNA concentrated at protrusive regions. Δ1 and Δ1 + 2 indicate deletions of the regions shown in (A). Scale bars: 10 μm.

C   Distribution of β-globin RNA or of *Ddr2* RNA detected in the same cells, quantified by measuring a Peripheral Distribution Index (PDI). *N* = 35–55 cells observed in 3 independent experiments. Bars: mean ± 95% CI. ****P < 0.0001 by analysis of variance with Dunnett's multiple comparisons test.

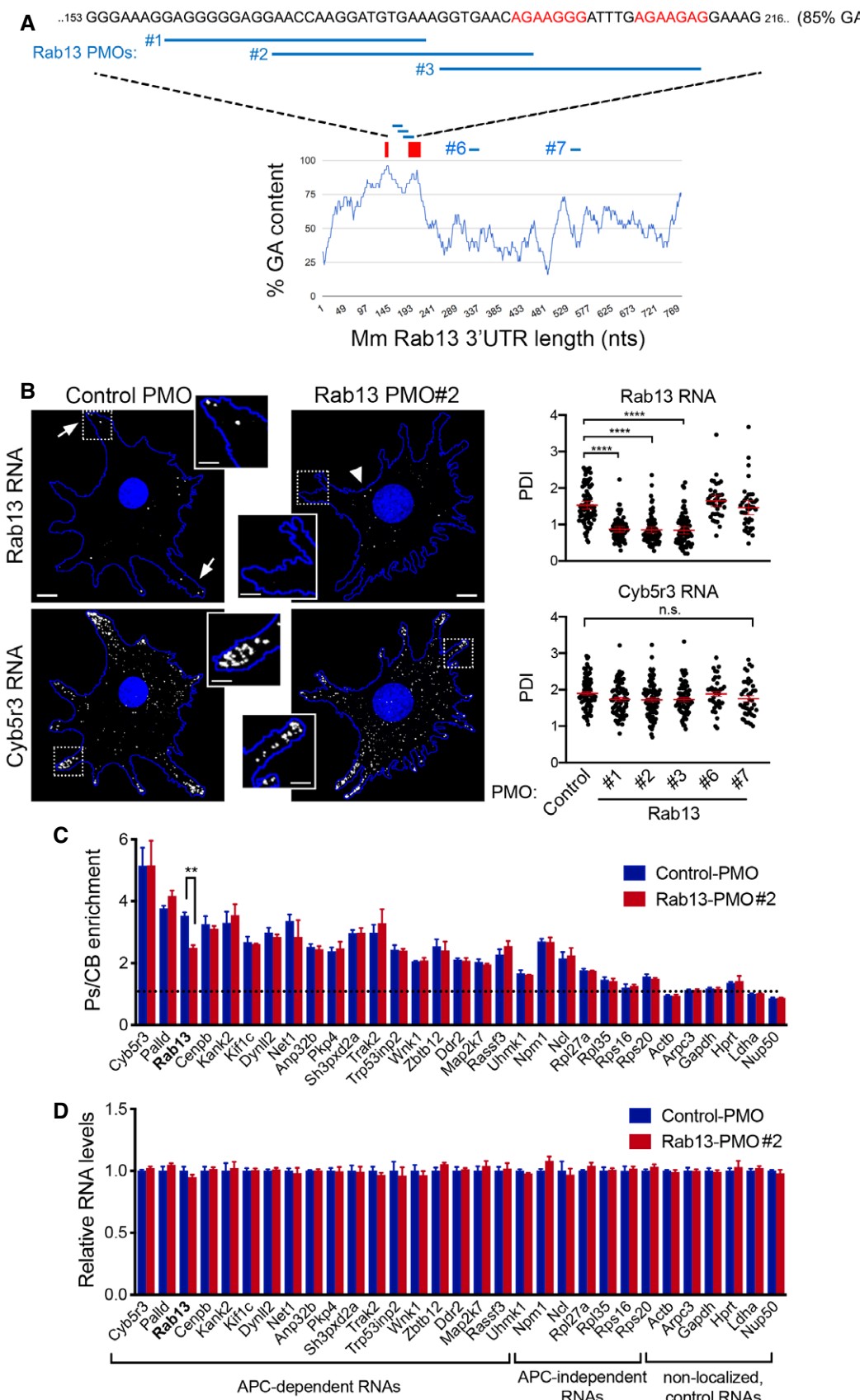

**Figure 3.**

◄

**Figure 3. Antisense oligonucleotides against the GA-rich region specifically interfere with localization of *Rab13* RNA.**

A  Schematic showing positions along the mouse *Rab13* 3′UTR targeted by the indicated PMOs. PMOs #1, 2, and 3 target the RGAAGRR motifs or the adjacent GA-rich region. PMOs #6 and #7 target the *Rab13* 3′UTR outside of the GA-rich region. The control PMO targets an intronic sequence of human β-globin. Red rectangles and text indicate the location of the GA-rich motifs.

B  FISH images and corresponding PDI measurements of mouse fibroblast cells treated with the indicated PMOs. *Cyb5r3* is an APC-dependent RNA also enriched at protrusions. Arrows: peripheral *Rab13* RNA. Arrowheads: perinuclear *Rab13* RNA. Boxed regions are magnified in the insets. Note that *Rab13* RNA becomes perinuclear in cells treated with PMOs against the GA-rich region. Scale bars: 10 μm. (4 μm in insets). ****$P < 0.0001$ by analysis of variance with Dunnett's multiple comparisons test. $N = 40$–90 cells observed in 3–6 independent experiments. Bars: mean ± 95% CI.

C  Protrusion (Ps) and cell body (CB) fractions were isolated from cells treated with control PMO or Rab13-PMO #2. The indicated RNAs were detected through nanoString analysis to calculate Ps/CB enrichment ratios ($n = 3$; bars: mean ± s.e.m.). Note that only the distribution of *Rab13* RNA is affected. **$P = 0.01$ by two-way ANOVA with Bonferroni's multiple comparisons test against the corresponding control.

D  Levels of the indicated RNAs were determined using nanoString analysis from control- or Rab13 PMO #2-treated cells ($n = 4$; bars: mean ± s.e.m.). No significant differences were detected by two-way ANOVA against the corresponding controls.

up by virtually all cells and persisted, either within apparent endosomal structures or released into the cytosol, for more than 3 days (Appendix Fig S1A and B). The effect of antisense PMOs on the localization of *Rab13* RNA was assessed, 3 days after PMO delivery, by single-molecule FISH of the endogenous *Rab13* RNA and PDI calculation. As expected, cells exposed to the control PMO exhibited peripheral localization of *Rab13* RNA. Similarly, the Rab13 #6 and #7 PMOs did not affect *Rab13* RNA distribution (Fig 3B). However, PMOs targeting the RGAAGRR motifs (PMOs #2 and #3) caused a pronounced mislocalization of *Rab13* RNA toward the perinuclear cytoplasm, evidenced by a significant reduction in PDI values (Fig 3B). Interestingly, the Rab13 PMO #1, which targets the adjacent GA-rich region, disrupted localization to a similar extent, suggesting that apart from the RGAAGRR motifs additional GA-rich sequences are important for localization or that the overall structure of this region is important. Notably, within the same cells, another APC-dependent RNA, *Cyb5r3*, which also contains GA-rich motifs, maintained its localization at protrusions under all conditions. Therefore, PMOs against the GA-rich region of *Rab13* RNA appear to specifically perturb *Rab13* RNA localization at protrusions.

To more extensively investigate the specificity of the observed effect, we assessed the distribution of a panel of ~ 20 APC-dependent RNAs, as well as of several APC-independent RNAs, using a protrusion/cell body fractionation scheme (Wang *et al*, 2017). As described previously (Wang *et al*, 2017), APC-dependent RNAs are enriched at protrusions, and their enrichment is more pronounced than that exhibited by APC-independent RNAs (Fig 3C). Importantly, cells treated with a mislocalizing Rab13 PMO exhibited indistinguishable distributions for all RNAs tested, with the notable exception of the targeted *Rab13* RNA, which became significantly less enriched at protrusions, corroborating and extending the FISH analysis described above (Fig 3B). We conclude that antisense PMOs against the Rab13 GA-rich region specifically alter the distribution of *Rab13* RNA without impacting the distribution of other RNAs, even those belonging to the same co-regulated group.

We additionally examined the overall abundance of the same panel of RNAs. PMO oligos do not trigger RNase H activity, and consistent with that, we did not observe any detectable change in the total levels of either *Rab13* or any other RNA, in cells treated with Rab13 PMOs (Fig 3D and Appendix Fig S2). Therefore, this approach allows us to specifically alter the distribution of the endogenous *Rab13* RNA without affecting its overall abundance in cells.

## The human RAB13 3′UTR exhibits a functionally conserved GA-rich region required for peripheral localization

*RAB13* RNA is localized at protrusive regions in diverse cell types and species. We thus sought whether similar sequence determinants support localization of the human *RAB13* transcript, which exhibits a GA-rich region (> 75%) and interspersed RGAAGRR motifs with similar topology as that of the mouse *Rab13* 3′UTR sequence (nts 98–268; Fig 4A). To identify functional regions with regard to RNA localization at protrusions, we delivered PMOs targeting regions across the length of the 3′UTR. PMOs targeting either RGAAGRR motifs directly (RAB13 PMOs 165 and 230) or adjacent GA-rich regions (RAB13 PMOs 91, 113, 191, and 210) (Fig 4A) significantly affected *RAB13* RNA localization. By contrast, all PMOs targeting sites outside of the GA-rich region did not affect RNA distribution (Fig 4A). Concomitant delivery of two individual PMOs (RAB13 PMOs 191 and 230) had an additive effect resulting in marked *RAB13* RNA mislocalization. Furthermore, the observed effects were specific for *RAB13* RNA since the distribution of another peripherally localized RNA, *NET1*, was not affected (Fig 4A and B). Thus, also in human cells, interfering with either the RGAAGRR motifs or the adjacent GA-rich regions specifically perturbs the peripheral localization of *RAB13* RNA.

Importantly, RNA mislocalization was not accompanied by any detectable change in the amount of RAB13 protein produced (Fig 4C). Specifically, the same amount of RAB13 protein is produced under basal conditions in cells exhibiting either peripheral (control PMO) or perinuclear (RAB13 PMO 230 or 191 + 230) *RAB13* RNA distribution (Fig 4C). This is consistent with the recently reported observation that *RAB13* RNA is similarly translated in both perinuclear and peripheral regions (Moissoglu *et al*, 2019). Furthermore, the location of the RAB13 RNA did not affect the total RAB13 protein levels or the amount of RAB13 protein at protrusions, upon acute stimulation with serum (Fig EV1). We conclude that, under both basal and stimulated conditions, the use of ASOs allows us to specifically assess the functional roles promoted by the localization of *RAB13* RNA without confounding contributions due to altered protein expression.

## Peripheral RAB13 RNA localization is important for cell migration

To understand the functional role of *RAB13* RNA localization at protrusions, we assessed the effect of *RAB13* RNA mislocalization

on the ability of cells to migrate, given that the encoded RAB13 protein promotes cell migration through multiple mechanisms (Ioannou & McPherson, 2016). For this, we compared cells treated with control PMOs or RAB13 mislocalizing PMOs using various assays. In one case, cells plated on microporous Transwell membrane inserts were induced to migrate toward a chemoattractant gradient and the number of cells reaching the bottom surface after 4 h was counted. Mislocalization of *RAB13* RNA from protrusions significantly perturbed the ability of cells to respond and migrate chemotactically (Fig 5A). To assess the migration speed of

individual cells, single cells expressing Cherry-NLS fluorescent protein to mark nuclei were plated on collagen and were tracked over time (Fig 5B and Movie EV1). Again, cells containing perinuclear *RAB13* RNA exhibited lower migration speeds (Fig 5B). Finally, cells attached on one side of a Matrigel plug were induced to migrate through it toward a chemoattractant in order to assess their ability to invade through a 3-dimensional matrix. Serial imaging sections through the Matrigel were taken for > 100 μm to assess the number of cells reaching at various depths. Notably, also in this case, cells treated with RAB13 mislocalizing PMOs exhibited

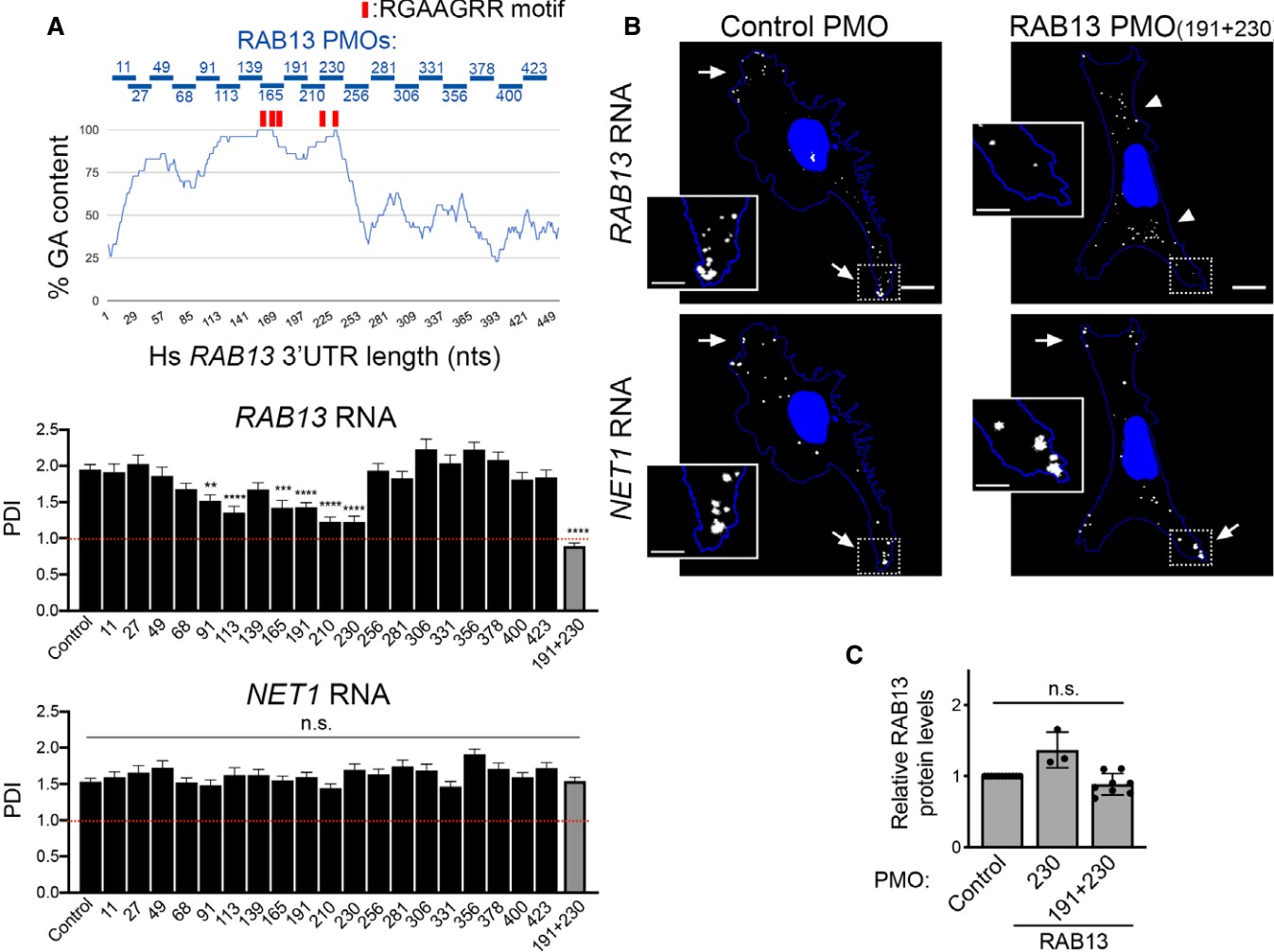

**Figure 4. The human *RAB13* 3′UTR contains a functionally conserved GA-rich region required for peripheral localization.**

A   Schematic showing %GA content and positions along the human *RAB13* 3′UTR targeted by the indicated PMOs. Red rectangles indicate the location of GA-rich motifs. Graphs present PDI measurements of *RAB13* RNA (upper panel) or *NET1* RNA (another APC-dependent RNA; bottom panel) detected in MDA-MB-231 cells treated with the indicated PMOs. PDI = 1 indicates a diffuse distribution. *P*-values: **< 0.01, ***< 0.001, ****< 0.0001 by analysis of variance with Dunnett's multiple comparisons test. n.s.: non-significant. *N* = 30–73 cells observed in 3–5 independent experiments. Bars: mean ± s.e.m.

B   Representative FISH images of cells treated with the indicated PMOs. Arrows point to peripheral RNA. Arrowheads point to perinuclear RNA. Boxed regions are magnified in the insets. Note that *RAB13* RNA becomes perinuclear in cells treated with PMOs against the GA-rich region, while *NET1* remains localized at protrusions. Scale bars: 10 μm (4 μm in insets).

C   RAB13 protein levels were measured by quantitative Western blot and normalized to total α-tubulin or GAPDH levels (for representative blot, see Fig 5E). Relative levels in RAB13 PMO-treated cells compared to control are shown. No significant differences were detected by Kruskal–Wallis test with Dunn's multiple comparisons test. *N* = 3–8. Bars: mean ± SD.

significantly reduced invasiveness (Fig 5C). Therefore, peripheral localization of *RAB13* RNA functionally contributes to various cell migration modes.

To assess in more detail the molecular mechanisms and polarized behaviors that underlie the observed migration defects, we examined the dynamics of protrusive or retractive cellular regions upon *RAB13* RNA mislocalization. Specifically, Lifeact-GFP-expressing cells were imaged over time and the average rate of extension or retraction of the cell edge was quantified, calculated from the velocity of each point along the cell boundary between successive movie frames (Barry *et al*, 2015; Fig 5D, Movies EV2 and EV3). Interestingly, mislocalization of *RAB13* RNA to perinuclear regions significantly reduced the rate of both protrusion and retraction (Fig 5D). Therefore, peripheral localization of *RAB13* RNA appears to impact overall the rate of cytoskeletal dynamics.

### Mislocalization of RAB13 RNA from the periphery phenocopies acute RAB13 protein loss

The cell migration defects observed upon mislocalization of *RAB13* RNA are quite remarkable given that the same amount of RAB13 protein is expressed in the cells (Fig 4C). We thus sought to determine the extent to which the overall RAB13 protein function is compromised when its RNA is prevented from reaching the periphery. For comparison, to set a baseline level, we knocked down RAB13 protein expression using siRNAs. Indeed, siRNA transfection reduced RAB13 protein to almost undetectable levels (Fig 5E). Consistent with prior reports (Ioannou *et al*, 2015), RAB13 knockdown resulted in reduced migration speed and compromised the rate of both protrusive and retracting motions (Fig 5E and F). Strikingly, the extent of these defects was very similar to that observed upon *RAB13* RNA mislocalization (Fig 5E and F). To determine whether the requirement for peripheral *RAB13* RNA stems from local production of RAB13 protein or whether the RNA might fulfill some unknown translation-independent role, we tried to rescue the observed migration defect by re-expression of wild-type GFP-RAB13

(carrying the full-length 3′UTR) or by re-expression of an identical construct carrying a frameshift point mutation (that introduces a termination codon at amino acid #10 of the RAB13 coding sequence) (Fig 5G and Appendix Fig S3). Importantly, rescue required the production of full-length RAB13 protein and could not be achieved when the frameshift mutation prevented RAB13 production from the same overall transcript (Fig 5G and Appendix Fig S3). Therefore, RAB13 protein is important for cell migration, but when it is translated at a perinuclear location (at the same level), the resulting protein is virtually non-functional toward cell migration, phenocopying the effect observed by an almost complete RAB13 protein loss.

### Peripheral RAB13 RNA translation is required for RAB13 protein activation but not steady-state distribution or membrane association

To understand this striking effect, we sought to determine how perinuclear versus peripheral translation affects the encoded RAB13 protein. To further confirm that any observed effects are due to changes in *RAB13* RNA distribution and not due to other potential non-specific consequences of PMO delivery, we additionally examined exogenously expressed GFP-RAB13 protein expressed from a construct that carries the full-length, wild-type RAB13 3′UTR or from a construct that carries the same UTR with a deletion of a 52-nt region that corresponds to the GA-rich region targeted by the mislocalizing RAB13 PMOs (ΔPMO UTR) (Fig EV3A). As predicted, FISH analysis showed that the exogenous wild-type *GFP-RAB13* RNA achieved a peripheral localization, while peripheral localization of the ΔPMO *GFP-RAB13* RNA was significantly abrogated and was similar to the localization of *GFP* RNA (Fig EV3A). Exogenous *GFP-RAB13* RNAs were expressed at similar levels (Fig EV3B) and did not affect the localization of another endogenous RNA (*NET1*; Fig EV3A). Furthermore, the amount of GFP-RAB13 protein produced from the two constructs was indistinguishable, as assessed by Western blot analysis or flow cytometry of GFP expression (Fig EV3C and D).

---

**Figure 5. Loss of peripheral *RAB13* RNA localization disrupts cell migration and phenocopies acute RAB13 protein loss.**

A  Transwell migration of MDA-MB-231 cells treated with control PMOs or RAB13 PMOs (191 + 230). Cells reaching the bottom surface after 4 h were counted. *n* = 25 fields of view in each of 6 independent experiments. Bars: mean ± s.e.m.

B  Cells expressing Cherry-NLS, and treated with the indicated PMOs, were tracked every 5 min for 10 h to derive average migration speed. *n* = 65 cells. Bars: mean ± s.e.m.

C  PMO-treated cells were induced to invade through a Matrigel plug. Cell staining intensity in arbitrary units (a.u.) was used to quantify relative invasion from *n* = 4 independent experiments. Bars: mean ± s.e.m.

D  Lifeact-GFP-expressing cells were treated with the indicated PMOs and imaged every minute over 1 h. Sequential image frames, from a control cell, highlight edge retraction (red arrowheads) or protrusion (yellow arrowheads). Corresponding edge velocity is shown, with negative values indicating retraction and positive values indicating extension. Average protrusion and retraction speeds were calculated from *n* = 11–13 cells. Bars: mean ± s.e.m. See also Movies EV2 and EV3.

E  Cells treated with the indicated PMOs or siRNAs were analyzed by Western blot (upper panels) to detect RAB13 and GAPDH protein levels. Migration speed was assessed as in (B) from *n* = 55–78 cells (bottom graph). Bars: mean ± s.e.m.

F  Protrusion and retraction speed of cells treated with the indicated PMOs or siRNAs were assessed as in (D). Graphs show average normalized values from *n* = 10–13 cells imaged for 1 h. Bars: mean ± s.e.m.

G  Migration speed of RAB13 knockdown cells re-expressing GFP-RAB13 or GFP-RAB13 with a frameshift (fs) mutation at the beginning of the RAB13 coding region. 28–52 cells were analyzed. Bars: mean ± s.e.m. Similar results were obtained in two additional independent experiments. Re-expressing cells were identified through GFP fluorescence, and cells with similar, low GFP signal were tracked. All cells are expressing cherry-NLS for accurate tracking.

Data information: *P*-values: *< 0.05; **< 0.01; ***< 0.001, ****< 0.0001, by Student's *t*-test (A–D), analysis of variance with Dunnett's multiple comparisons test (E, F) or Kruskal–Wallis test with Dunn's multiple comparisons test (G). n.s.: not significant.
Source data are available online for this figure.

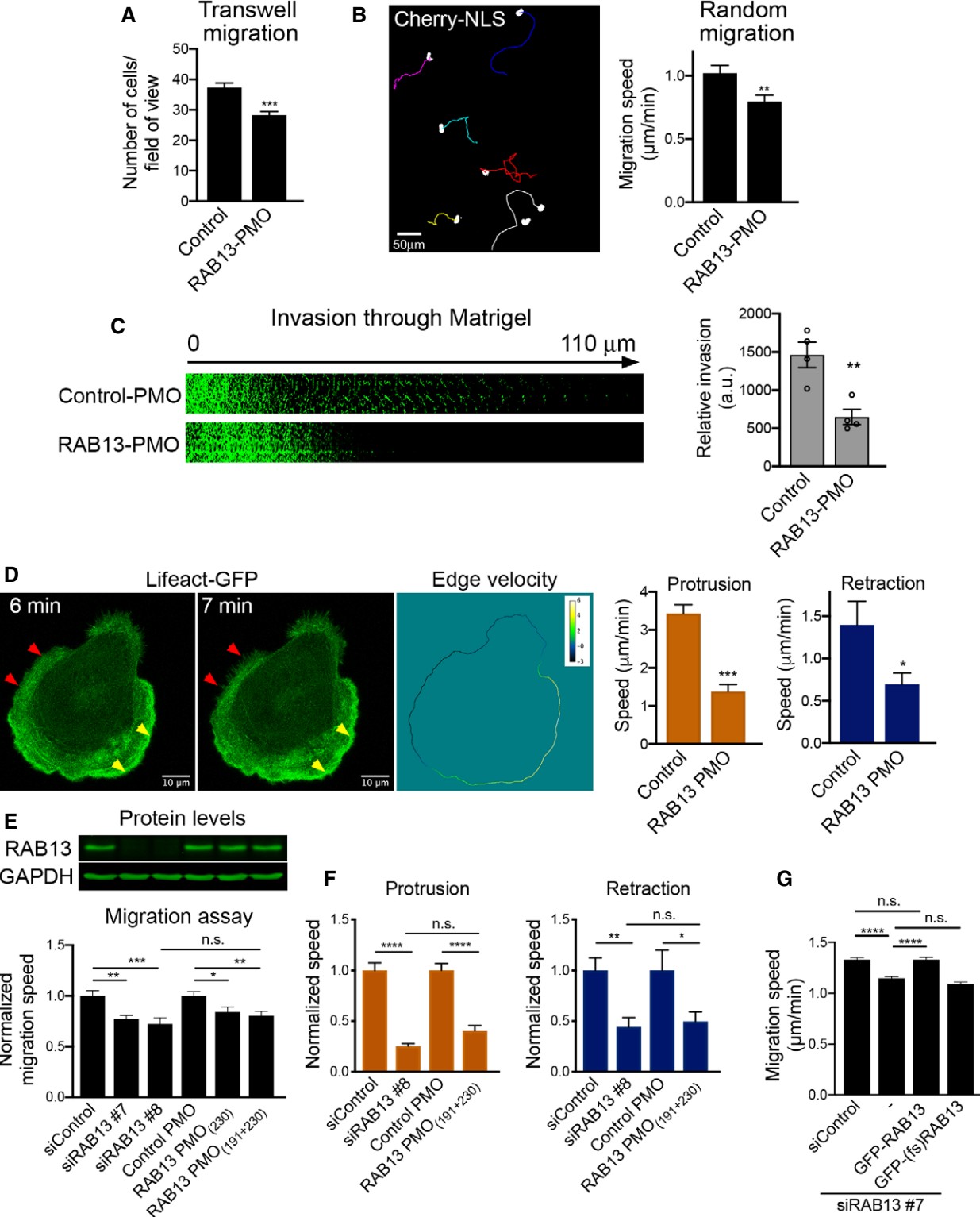

**Figure 5.**

Interestingly, the steady-state localization of RAB13 protein was not impacted upon changing the location of its translation. Endogenous RAB13 protein assumed the same membranous and perinuclear distribution upon PMO delivery (Fig 6A). Similarly, exogenous GFP-RAB13 showed identical distribution regardless of where its RNA was localized (Fig 6B). Because Rab GTPases associate with membranes to control vesicle trafficking, we further examined whether association of RAB13 with membranes was affected by the

location of translation. For this, we permeabilized cells with digitonin and isolated soluble and particulate fractions. Again, we found that both endogenous and exogenous RAB13 protein associated similarly with the particulate fraction regardless of where the encoding RNA was enriched (Figs 6C and EV4A). Furthermore, consistent with the observed similar association with the membrane fraction, RAB13 protein translated peripherally or perinuclearly associates to the same extent with the Rab escort protein REP-1 (Fig EV4B), an interaction required for prenylation of Rab GTPases (Leung *et al*, 2006a), as well as with RabGDI (Fig EV4B), a protein that extracts Rab GTPases from membranes and maintains them soluble in the cytoplasm (Muller & Goody, 2018). Therefore, despite the pronounced functional defect exhibited upon *RAB13* RNA mislocalization, at least some of the properties and interactions of RAB13 protein are not affected.

We then investigated whether perinuclear translation affects RAB13 activity. Active Rab proteins are loaded with GTP through the action of guanine nucleotide exchange factors (GEFs). GTP loading promotes Rab binding to effector molecules. We assessed RAB13 activity through measuring its ability to interact with the RAB13-binding domain (RBD) of its effector protein MICAL-L1 (Ioannou *et al*, 2015). Interestingly, endogenous RAB13 produced from a perinuclear RNA (RAB13 PMO cells) exhibited significantly reduced activity compared to peripherally translated RAB13 protein (control PMO cells or cells treated with PMOs against the unrelated *NET1* RNA) (Fig 6D). In agreement, pull-down assays with the MICAL-L1 RBD also revealed that the exogenous GFP-RAB13 protein expressed from a perinuclearly localized RNA had significantly less activity compared to peripherally translated protein (GFP-RAB13 with wtUTR) (Fig 6E). Therefore, RAB13 activity and function in cell migration are determined by the subcellular location of *RAB13* RNA translation.

### Peripheral RAB13 RNA promotes the local association of RAB13 with the exchange factor RABIF

To understand the underlying mechanism, we used mass spectrometry analysis to identify RAB13-interacting proteins. Specifically, we identified proteins bound to wild-type GFP-RAB13 or GFP-RAB13 (T22N), a nucleotide-free mutant which binds tightly to putative GEFs (Table EV1). As expected, wild-type GFP-RAB13 is associated with RabGDI isoforms. The GFP-RAB13(T22N) mutant did not associate with RabGDI but interacted with RABIF/MSS4, a protein known as a GEF for Rab GTPases (Burton *et al*, 1993, 1994; Miyazaki *et al*, 1994; Table EV1). Western blot analysis verified the association of RABIF with RAB13 and revealed that it interacts preferentially with RAB13(T22N), as would be expected from a GEF (Fig 7A and B). To confirm that RABIF acts as a RAB13 GEF in our system, we used CRISPR/Cas9 to knockout its expression (Fig 7C). Indeed, two different guide RNAs targeting RABIF led to a significant reduction of RAB13 activation levels (Fig 7C). Furthermore, importantly, expression of RAB13(T22N) from a perinuclear RNA (carrying the ΔPMO UTR) led to a significantly reduced association with RABIF (Fig 7A and B), strongly suggesting that RABIF is part of the mechanism linking *RAB13* RNA localization to RAB13 protein activation.

The above data show that peripheral localization of *RAB13* RNA affects the degree of RAB13-RABIF interaction. However, both

proteins exhibit a steady-state perinuclear enrichment (Figs 1B and EV5A). To understand this effect, we examined whether this interaction occurs in a spatially specific manner. For this, we used proximity ligation assay (PLA) to detect protein–protein interactions *in situ*. PLA between endogenous RAB13 and RABIF revealed a specific interaction between the two, since the signal was significantly decreased upon knockdown of each binding partner (Fig 7D and Appendix Fig S4A). RAB13-RABIF interaction was detected in both peripheral and perinuclear regions. However, interestingly, mislocalization of *RAB13* RNA upon PMO delivery reduced specifically the peripheral RAB13-RABIF interaction (Fig 7E and Appendix Fig S4B). Similarly, interaction between GFP-RAB13 (T22N) and RABIF occurred in both peripheral and perinuclear regions and expression of GFP-RAB13 from a mislocalized construct (carrying the ΔPMO UTR) specifically reduced the observed peripheral interaction (Fig 7F). Thus, a peripheral pool of RAB13-RABIF complex requires peripheral localization of *RAB13* RNA.

### RABIF associates with RAB13 co-translationally

The above data suggest that RABIF might interact with RAB13 co-translationally and thereby, where *RAB13* RNA is located and translated, determines the site of complex formation. To determine whether a co-translational interaction occurs, we immunoprecipitated RABIF and tested for the presence of *GFP-RAB13* RNA (Fig 8A). Indeed, both *GFP-RAB13* and *GFP-RAB13(T22N)* RNAs were readily detected above background in RABIF immunoprecipitates (Fig 8B). To further assess whether this interaction depends on active translation, immunoprecipitations were carried out from cells pre-treated with puromycin, an antibiotic which dissociates translating ribosomes. Importantly, puromycin treatment abolished the binding of RABIF to *RAB13* RNA (Fig 8B). These data indicate that RABIF does not directly associate with *RAB13* RNA and argue against a model whereby the UTR scaffolds interactions with the newly synthesized polypeptide (Berkovits & Mayr, 2015; Lee & Mayr, 2019). We rather interpret these data to indicate that RABIF co-translationally binds to the nascent RAB13 protein.

## Discussion

We show here that, for the protrusion-localized *RAB13* RNA, peripheral translation does not lead to a corresponding local protein accumulation. Rather the location of translation critically affects some, but not all properties of the encoded RAB13 protein. Specifically, the location of translation does not affect the steady-state RAB13 protein distribution, its membrane association, or interaction with Rab regulators such as REP-1 and RabGDI. Strikingly however, translation at the periphery is required for full RAB13 activation, likely through favoring a co-translational association with the RABIF GEF (Fig 8C). Preventing peripheral translation specifically reduces the peripheral RAB13-RABIF binding and compromises the ability of RAB13 to support cell migration to an extent similar to that observed upon almost complete loss of the protein. This work identifies the subcellular location of *RAB13* RNA translation as a critical factor determining the exact functionality and properties of the encoded protein.

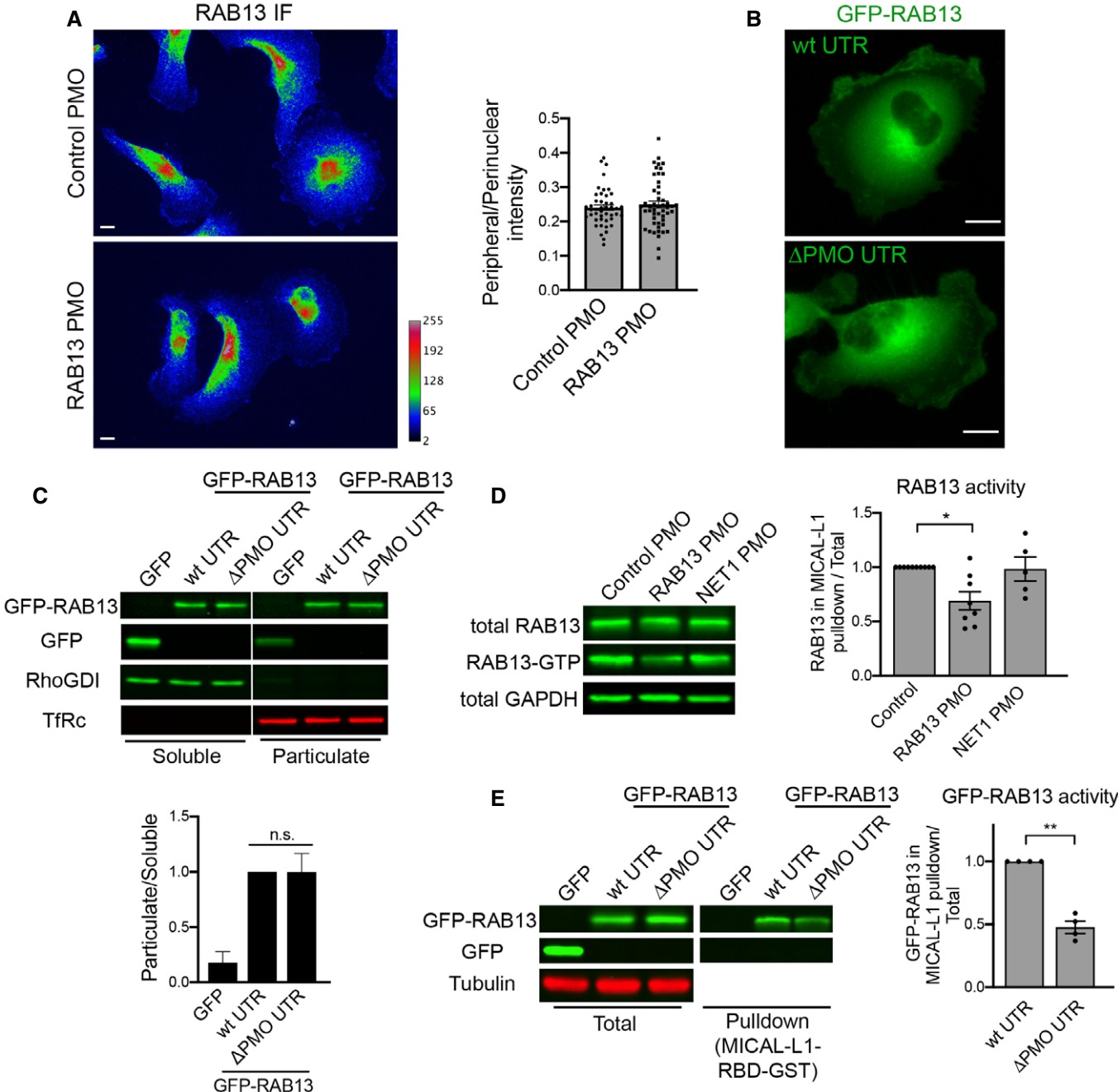

**Figure 6. Peripheral *RAB13* RNA translation is required for RAB13 protein activation but not steady-state distribution or membrane association.**

A   Wide-field images of RAB13 immunofluorescence in MDA-MB-231 cells treated with control or RAB13 (191 + 230) PMOs and ratios of peripheral/perinuclear intensity. Scale bars: 10 μm. n = 45–50 cells. Bars: mean ± s.e.m. Similar results were observed in two additional independent experiments.

B   Fluorescence images (projections of confocal slices throughout the cell height) of cells expressing GFP-RAB13 with the indicated UTRs. Note that in both cases the protein assumes indistinguishable distribution. Scale bars: 10 μm.

C   Soluble/particulate fractionation of the indicated cell lines followed by Western blot to detect the indicated proteins. RhoGDI and TfRc serve as soluble and particulate markers, respectively. Graph shows quantitation from n = 3 independent experiments. Bars: mean ± s.e.m.

D, E   Active RAB13 (RAB13-GTP) was pulled down using MICAL-L1 RBD-GST from the indicated PMO-treated cells (D) or GFP-RAB13-expressing lines (E). The amount of endogenous or exogenous RAB13 was measured by quantitative Western blot, and relative levels of active RAB13 are plotted. n = 8 (D), n = 4 (E). Bars: mean ± s.e.m.

Data information: *P*-values: *< 0.05, **< 0.01 by Kruskal–Wallis test.
Source data are available online for this figure.

It is increasingly appreciated that the process of translation can affect the encoded proteins through various mechanisms. For instance, the translation machinery provides a platform for recruitment of maturation factors that guide nascent polypeptide chains into functional protein structures (Gloge *et al*, 2014). Furthermore, subunits of hetero-oligomeric complexes come into contact and begin assembling co-translationally (Shiber *et al*, 2018). Such co-translational interactions can determine the efficiency of functional complex formation (Shieh *et al*, 2015). Additionally, the

untranslated regions (UTRs) of the RNA template can impact the fate of the newly synthesized proteins by acting as scaffolds to recruit protein complexes that affect the targeting or modification of the nascent or newly synthesized polypeptides (Basu *et al*, 2011; Berkovits & Mayr, 2015). In this way, use of alternative UTRs can direct formation of different protein complexes that support distinct downstream protein functions (Berkovits & Mayr, 2015; Lee & Mayr, 2019). Our results suggest that another factor that can have a determining impact on downstream protein activity

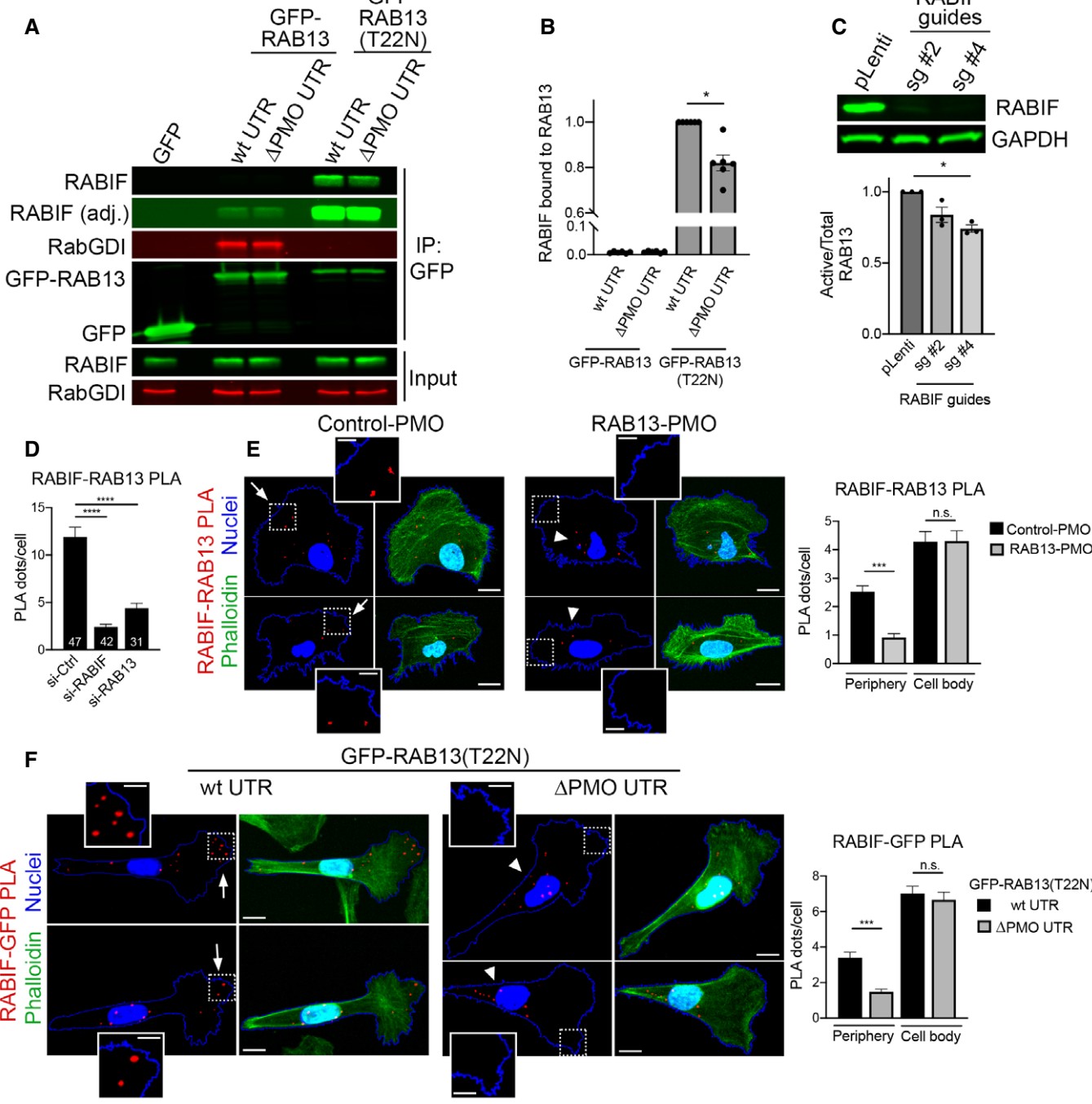

Figure 7.

**Figure 7. Peripheral *RAB13* RNA promotes the local association of RAB13 with the exchange factor RABIF.**

A   Immunoprecipitation and Western blot analysis to detect proteins bound to GFP-RAB13 expressed from constructs carrying wt or ΔPMO RAB13 3′UTR. GFP-RAB13 (T22N) expresses a nucleotide-free mutant which binds tightly to putative GEFs. RABIF panel is also shown with adjusted contrast to reveal lower binding to wtGFP-RAB13.

B   Quantification of RABIF binding to RAB13 from experiments as in (A). *n* = 6. Bars: mean ± s.e.m.

C   Active RAB13 pull-down assay from cells with CRISPR knockdown of RABIF using the indicated sgRNAs. *n* = 3. Bars: mean ± s.e.m.

D   Quantification of RABIF-RAB13 PLA signal from cells transfected with the indicated siRNAs. Number of observed cells is indicated within each bar. Bars: mean ± s.e.m. Similar results were observed in one additional independent experiment.

E   Representative RABIF-RAB13 PLA images and quantitations from cells transfected with the indicated PMOs. *N* > 70 cells. Bars: mean ± s.e.m. Arrows indicate PLA signal at the cell periphery. Arrowheads indicate PLA signal within the cell body. Boxed regions are magnified in the insets. Scale bars: 15 μm; 5 μm in insets.

F   Representative RABIF-GFP PLA images and quantitations from cells expressing GFP-RAB13(T22N) carrying wt or ΔPMO RAB13 3′UTR. *N* > 70 cells. Bars: mean ± s.e.m. Arrows indicate PLA signal at the cell periphery. Arrowheads indicate PLA signal within the cell body. Boxed regions are magnified in the insets. Scale bars: 10 μm; 4 μm in insets.

Data information: *P*-values: *< 0.05, **< 0.01, ***< 0.001, ****< 0.0001 by Wilcoxon matched-pairs signed-rank test (B) or analysis of variance with Dunn's, Dunnett's, or Tukey's multiple comparisons test (C–F).

Source data are available online for this figure.

is the subcellular microenvironment into which a protein is being synthesized.

Peripheral translation affects RAB13 function without resulting in a corresponding accumulation of RAB13 at the periphery. This finding contrasts with observations, in yeast or mammalian neurons, of other localized RNAs whose translation leads to local increase in protein concentration (Paquin & Chartrand, 2008; Rangaraju *et al*, 2017; Holt *et al*, 2019). A common feature of these latter cases is that premature or ectopic translation would have deleterious consequences, such as disruption of cell type identity in yeast, or incorrect axonal pathfinding or synaptic marking in neuronal cells. We believe that another relevant point is that such examples of localized RNA transcripts, for the most part, pertain to RNAs which are localized in rather stably polarized structures, whose identity remains fixed even though specific signals might elicit transient responses. By contrast, protrusions of migrating cells are very dynamic switching between extension and retraction phases as they explore the surrounding environment (Ryan *et al*, 2012). Directional migration requires that cells are able to quickly repolarize and redefine their leading and retracting areas to follow the direction of an asymmetric extracellular cue (Petrie *et al*, 2009). Given that such dynamic changes can occur within minutes, which is the same range as the time required for translation of an average size protein, it seems unlikely that the use of local translation as a means of building dynamic local concentration gradients would be an efficient process. In light of the evidence presented here, we rather propose that in dynamically changing protrusions, local translation serves to expose newly synthesized proteins to the local environment and thus impart on them particular properties that might persist even after they have diffused or trafficked away from their site of synthesis. Given that global studies have shown little correlation between RNAs localized at cell protrusions and steady-state concentration of the corresponding proteins (Mardakheh *et al*, 2015), it would appear that this mode of regulation might be relevant for the majority of protrusion-localized RNAs.

In the case of RAB13, peripheral translation is required for its association with RABIF in this local environment. Our data indicate that RABIF and RAB13 associate co-translationally and that their interaction at the periphery is required for full RAB13 activation and RAB13 function in cell migration. We propose that this peripherally activated RAB13 pool is specifically directed toward migration-relevant effectors, since mislocalization of *RAB13* RNA phenocopies complete loss of protein expression with regard to cell migration (Fig 8C). Interestingly, this occurs even though a substantial amount of GTP-loaded, membrane-bound RAB13 can still be detected. Therefore, RAB13 protein, that is not produced peripherally, can be activated and likely contributes to different cellular functions, perhaps by interacting with different effectors. Indeed, RAB13 has been shown to participate in diverse trafficking events including the recycling of integrins (Wu *et al*, 2011; Nishikimi *et al*, 2014; Ioannou *et al*, 2015; Sahgal *et al*, 2019) or of factors involved in cell–cell contacts (Marzesco *et al*, 2002; Kohler *et al*, 2004; Morimoto *et al*, 2005; Terai *et al*, 2006). Given that significant RAB13 activity remains in the absence of RABIF, RAB13 activation additionally relies on other GEFs. These could include proteins of the DENND family, such as DENND2B and DENND1C (Nishikimi *et al*, 2014; Ioannou *et al*, 2015) even though we have not been able to detect expression of DENND2B in MDA-MB-231 cells. RABIF additionally exhibits chaperone activity (Gulbranson *et al*, 2017), and, consistent with that, we observe a reduction of RAB13 levels upon RABIF knockout (Fig EV5B), but not upon *RAB13* RNA mislocalization. We propose that RABIF affects RAB13 differently based on the particular subcellular environment, acting as a GEF at the periphery and as a chaperone in the cell body (Fig 8C). What distinguishes these complexes on a molecular level is not clear; however, potential candidates could involve spatially restricted post-translational modifications occurring on RAB13 and/or RABIF. Our data do not rule out that perinuclear RAB13-RABIF interaction also occurs co-translationally (Fig 8C). Understanding whether this is the case or not could help narrow down on more detailed mechanistic possibilities.

Intriguingly, even though RAB13 translation occurs primarily at extending protrusions (Moissoglu *et al*, 2019), *RAB13* RNA mislocalization affects both extension and retraction dynamics (Fig 5D and F). We think likely that the effect on retraction is an indirect manifestation of feedback mechanisms which coordinate extension and retraction during cell movement (Ridley *et al*, 2003). However, alternative possibilities could be envisioned including, for example, that a reduced stoichiometry of newly synthesized RAB13 at the back might signal edge retraction or that the formation of *RAB13* RNA clusters at retracting tails (Moissoglu *et al*, 2019) might themselves play a role in retraction.

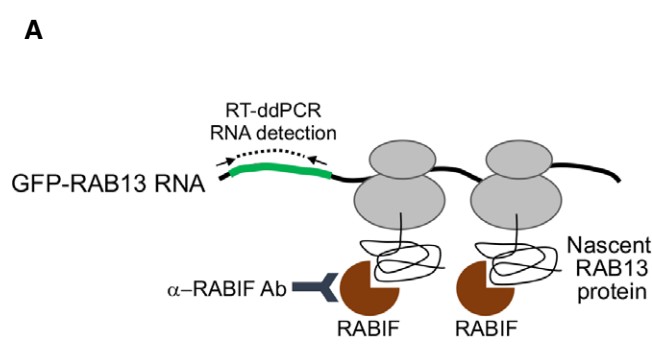

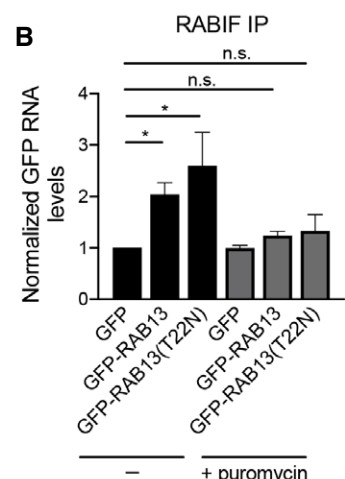

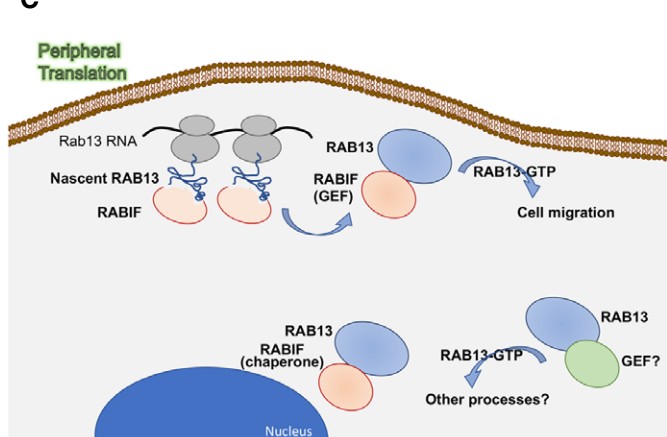

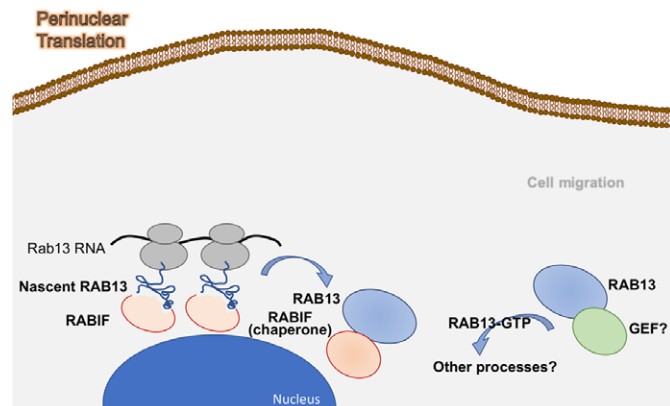

**Figure 8. RABIF associates with RAB13 co-translationally.**

A Schematic depicting experimental strategy for assessing co-translational association of RABIF with nascent RAB13.

B Quantification of GFP RNA levels associating with RABIF in immunoprecipitation assays from the indicated cell lines. Note that even though RABIF binds several-fold more to RAB13(T22N) (see Fig 7A and B), it binds similarly to the wild-type and T22N *RAB13* RNA, indicating that RABIF binds similarly to nascent RAB13. After translation, it is likely displaced upon GTP loading of wild-type RAB13, while it remains more stably bound to the nucleotide-free (T22N) form. N = 6. Bars: mean ± s.e.m. P-values: *< 0.05 by analysis of variance with Dunn's multiple comparisons test.

C Proposed model for regulation of RAB13 activity through local RNA translation. See text for details.

Overall, our data highlight a role for local translation in directing Rab GTPase activation and suggest that the microenvironment into which a protein is synthesized can affect its regulation and downstream function. Concurrent work by Costa *et al* (2020) has identified a similar GA-rich region involved in localization of the RAB13 RNA in endothelial cells and has demonstrated its importance during zebrafish blood vessel morphogenesis, further supporting the generality and physiological relevance of the mechanism described here.

## Materials and Methods

### Plasmid constructs and lentivirus production

To express β-globin RNA followed by different UTRs, the genomic region containing the coding sequence of human β-globin was ligated to fragments corresponding to the following UTRs: β-globin UTR (3′UTR of human β-globin, Accession#: NM_000518.4), Rab13 UTR (wt) (3′UTR of mouse Rab13, Accession#: NM_026677.4), Rab13 UTR (Δ1) (3′UTR of mouse Rab13 missing nucleotides 204–211, corresponding to region 1), and Rab13 UTR (Δ1 + 2) (3′UTR of mouse Rab13 missing nucleotides 204–211 and 193–200, corresponding to regions 1 and 2, respectively). The sequences were cloned into the pENTR1A vector and then transferred into the pINDUCER20 lentivector (Addgene #44012) using the Gateway LR Clonase II Enzyme Mix (Thermo Fisher Scientific, cat# 11791-020).

mEGFP-Lifeact-7 (gift of Michael Davidson; Addgene plasmid # 54610) was transferred into pCDH-CMV-MCS-EF1-Puro (System Biosciences, cat #CD510B-1) using NheI/NotI sites for virus production.

To express N-terminally GFP-tagged RAB13 using different UTRs, the coding sequence of EGFP was ligated in frame to the coding

sequence of human RAB13. The fusion was then ligated to fragments corresponding to the 3′UTR of human RAB13 (Accession#: NM_002870.5) (wt UTR) or the 3′UTR of RAB13 missing 53 nucleotides (202–254) from the GA-rich region that is targeted by the RAB13 PMOs (ΔPMO UTR). The sequences were cloned into PCDH-PGK lentivector.

For rescue experiments, the coding sequence and 3′UTR of mouse Rab13 (Accession#: NM_026677.4) were cloned into pEGFP-C1 vector to generate GFP-Rab13/wt UTR. Note that the mouse Rab13 sequence is not targeted by the RAB13 siRNA #7 used. To generate GFP-(fs)Rab13/wt UTR, site-directed mutagenesis was used to introduce a T residue, creating a stop codon, at position 28 of the Rab13 coding sequence.

Lentivirus was produced in HEK293T cells cultured in DMEM containing 10% FBS and penicillin/streptomycin. HEK293T cells were transfected with pINDUCER20 lentivectors, together with packaging plasmids pMD2.G and psPAX2 using PolyJet In Vitro DNA Transfection Reagent (SignaGen) for 48 h. Harvested virus was precipitated with polyethylene glycol at 4°C overnight.

## Immunofluorescence and Western blot

For IF, cells were plated onto collagen IV-coated glass coverslips (10 µg/ml) and then fixed with 4% paraformaldehyde (PFA) in PBS (phosphate-buffered saline) for 15 min, permeabilized with 0.2% Triton X-100 in PBS for 5 min, blocked with 5% goat serum in PBS for 1 h, and incubated with Rab13 rabbit polyclonal antibody (Novus Biologicals, cat# NBP1-85799, 1/200 dilution) for 1 h. Secondary antibodies were conjugated with Alexa 647 (Thermo Fisher Scientific).

For Western blot detection, the following antibodies were used: anti-Rab13 rabbit polyclonal (Novus Biologicals, cat# NBP1-85799, 1/1,000 dilution), anti-GFP rabbit polyclonal (Invitrogen, cat# A11122, 1/2,000 dilution), anti-RhoGDI, anti-transferrin receptor (clone H68.4, Thermo Fisher Scientific, cat# 13-6800), anti-α-tubulin mouse monoclonal (Sigma-Aldrich, cat# T6199, 1/10,000 dilution), anti-GAPDH rabbit monoclonal 14C10 (Cell Signaling Technology, cat# 2118, 1/2,000 dilution), anti-phospho-FAK(Y397) (Cell Signaling Technology, cat# 3283), anti-REP-1 (Abcam, cat# 134964), anti-RabGDI (Santa Cruz Biotechnology, cat# sc-374649), and anti-RABIF mouse monoclonal antibody D-12 (Santa Cruz biotechnology, cat# sc-390759, 1/1,000 dilution).

## Cell culture, siRNA transfection, and generation of stable cell lines

NIH/3T3 mouse fibroblast cells (ATCC) were grown in DMEM supplemented with 10% calf serum, sodium pyruvate, and penicillin/streptomycin (Invitrogen) at 37°C, 5% CO$_2$. MDA-MB-231 human breast cancer cells (ATCC) were grown in Leibovitz's L15 media supplemented with 10% fetal bovine serum (FBS) and penicillin/streptomycin at 37°C in atmospheric air. For serum stimulation, MDA-MB-231 cells were serum-starved overnight in the presence of 0.1% FBS and stimulated for the indicated times by addition of 10% FBS. Cells have been tested for mycoplasma and are free of contamination.

For knockdown experiments, 40 pmoles of siRNAs were transfected into cells with Lipofectamine RNAiMAX (Thermo Fisher

Scientific, cat# 13778-150) according to the manufacturer's instructions. Cells were assayed after 3 days. siRNAs used were as follows: AllStars negative control siRNA (cat# 1027281), si-Rab13 #7 (cat# SI02662149; target sequence: 5′-CAGGGCAAACATAAATGTAAA-3′), si-Rab13 #8 (cat# SI02662702; target sequence: 5′-ATGGTCT TTCTTGGTATTAAA-3′), and si-RABIF (Cat# SI00301595) from Qiagen. For rescue experiments, cherry-NLS-expressing cells were transfected with siRNAs. Thirty-six hours after siRNA transfection, the cells were transfected with the indicated GFP-RAB13 constructs using Lipofectamine LTX (Thermo Fisher Scientific, cat# A12621) and GFP-positive cells were assayed for migration 24 h later.

To generate stable cell lines expressing β-globin reporters with different UTRs, NIH/3T3 cells were infected with the corresponding lentiviruses and selected with 0.6 mg/ml geneticin (Thermo Fisher Scientific). Exogenous expression of the reporters was induced by addition of 1 µg/ml doxycycline (Fisher Scientific) approximately 2–3 h before processing cells for FISH. This concentration of doxycycline and duration of incubation were chosen to achieve relatively low levels of expression and prevent competition effects.

To generate stable cell lines expressing GFP-RAB13 with different UTRs, MDA-MB-231 cells were infected with the corresponding lentiviruses and selected with 6 µg/ml blasticidin (Thermo Fisher Scientific). GFP-expressing cells with low level of GFP expression were sorted by FACS.

To generate stable RABIF knockout cell lines using CRISPR-Cas9 technology, sgRNAs targeting RABIF (sg #2: 5′-GAGCGAGT-TAGTGTCAGCCGAGG-3′ and sg #4 5′-AGCGAGTTAGTGTCAGCC-GAGGG-3′) were cloned in lentiCRISPRv2 (Addgene #52961). MDA-MB-231 cells were infected with the corresponding lentiviruses and selected with 1 µg/ml puromycin (Thermo Fisher Scientific).

## Morpholino ASOs and delivery

PMOs were synthesized by Gene Tools, LLC, and delivered into cells using Endo-Porter (PEG) (Gene Tools, LLC). Sequences used are listed in Table EV2.

## Protrusion/cell body isolation and RNA analysis

Protrusions and cell bodies were isolated from serum-starved cells plated for 2 h on Transwell inserts equipped with 3.0-µm porous polycarbonate membrane (Corning) as previously described (Wang *et al*, 2017) with some modifications. Briefly, 3.5 million serum-starved cells were plated per 25-mm filter and pre-coated with 10 µg/ml collagen IV, and 1 or 3 filters were used for cell body or protrusion isolation, respectively. Cells were allowed to attach for 2 h, and LPA (150 ng/ml) was added to the bottom chamber for 40 min. The cells were fixed with 0.3% paraformaldehyde for 10 min. For isolating protrusions, cell bodies on the upper surface were manually removed by wiping with cotton swab and laboratory paper. The protrusions on the underside were then solubilized by immersing the filter in crosslink reversal buffer (100 mM Tris pH 6.8, 5 mM EDTA, 10 mM DTT, and 1% SDS) and gentle scraping. Cell bodies were similarly isolated after manually removing protrusions from the underside of the membrane. The extracts were incubated at 70°C for 45 min and used for protein analysis or RNA isolation using TRIzol LS (Thermo Fisher Scientific). Note that for

these experiments a cell line stably expressing GFP was used in order to use the GFP RNA/protein as an additional internal control.

For nanoString analysis, RNA samples were analyzed using a custom-made codeset and the nCounter analysis system according to the manufacturer's instructions (NanoString Technologies). For ddPCR, RNA samples were analyzed using the ddPCR EvaGreen Supermix (Bio-Rad, cat. no. 186-4034). Droplets were generated using the Automated Droplet Generator (Bio-Rad, cat no. 186-4101), PCR amplification was performed on a C1000 Touch™ Thermal Cycler (Bio-Rad, cat no. 185-1197), and droplet reading was done with the QX-200 Droplet Reader (Bio-Rad, cat no. 186-4003) and QuantaSoft software (Bio-Rad).

## Fluorescence *in situ* hybridization (FISH)

For FISH, cells were plated on fibronectin- or collagen IV-coated glass coverslips for 2–3 h and subsequently fixed with 4% PFA for 20 min (5 µg/ml fibronectin for NIH/3T3 cells and 10 µg/ml collagen IV for MDA-MB-231 cells). FISH was performed with ViewRNA ISH Cell Assay Kit (Thermo Fisher Scientific) according to the manufacturer's instructions. The following probe sets were used: human HBB #VA1-13382; mouse Ddr2 #VB6-12897; mouse Rab13 #VB1-14374; mouse Cyb5r3 #VB6-3197970; human Rab13 #VA1-12225; human Net1 #VA6-3169338; and GFP #VF6-16198. DAPI was used to stain nuclei, and CellMask stain (Thermo Fisher Scientific) was used to identify the cell outlines. Samples were mounted with ProLong Gold Antifade Reagent (Thermo Fisher Scientific). Image analysis and quantification of RNA distributions were performed using RDI Calculator (Stueland *et al*, 2019).

## Active RAB13 pull-down assays, cell fractionation, and immunoprecipitation

For RAB13 pull-down, cells were washed with ice-cold TBS and lysed in buffer containing 50 mM Tris (pH 7.4), 150 mM NaCl, 10 mM MgCl$_2$, 1% Triton X-100, 0.1% SDS, 0.5% sodium deoxycholate, 10% glycerol, and protease inhibitor cocktail. Clarified lysates were then incubated with 20 µg of recombinant glutathione *S*-transferase (GST)-RBD-MICAL-L1 and Pierce™ Glutathione Magnetic Agarose Beads (Life Technologies, Invitrogen, cat# 78601) for 2 h at 4°C. Beads were washed with lysis buffer without sodium deoxycholate and SDS, eluted with sample buffer, and analyzed by immunoblotting.

To obtain soluble and particulate fractions, adherent cells were incubated with a buffer containing 50 mM Tris pH 7.4, 50 mM NaCl, 0.01% digitonin, 10 mM MgCl$_2$, 1 mM EDTA, 10% glycerol, and a cocktail of protease and phosphatase inhibitors (Thermo Fisher Scientific, cat# 1861281) with rocking at 4°C for 10 min. Cell material extracted in this buffer was defined as soluble. The remaining material, corresponding to the particulate fraction, was extracted using a buffer containing 50 mM Tris pH 7.4, 1% Triton X-100, 150 mM NaCl, 10 mM MgCl$_2$, 0.1% SDS, 0.5% deoxycholate, and 10% glycerol. Equal volumes of these fractions were analyzed by SDS–PAGE and immunoblotting. For quantitation, the soluble and particulate amounts were normalized to the corresponding amounts of soluble and particulate markers.

For immunoprecipitations, cells were lysed with a buffer containing 50 mM Tris pH 7.4, 1% NP-40, 150 mM NaCl, 10 mM MgCl$_2$,

10% glycerol, and a cocktail of protease and phosphatase inhibitors (Thermo Fisher Scientific, cat# 1861281). Lysates were cleared by centrifugation and mixed with GFP-Trap Magnetic Agarose Beads (Chromotek, cat# gtma-10) or Protein G Dynabeads (Fisher Scientific) for 1.5 h at 4°C. Immobilized complexes were eluted with Laemmli's buffer and analyzed by SDS–PAGE and immunoblotting.

## Migration assays

Transwell migration assays were performed using 24-well plates with 8-µm pore-size inserts. Inserts were coated on both sides with collagen IV overnight (10 µg/ml in PBS). MDA-MB-231 cells that had been kept in 0.1% FBS-containing media for 16 h were detached using trypsin. After addition of trypsin inhibitor, cells were resuspended in media containing 0.1% FBS and 20,000 cells (in 400 µl) were plated onto each insert. The lower chamber contained 600 µl of the same media. Two hours post-plating, 10% FBS was added in the lower chamber to induce cell motility. Two hours later, both sides of the membrane were fixed with 4% PFA, cells were removed from the upper chamber, and the membrane was stained with DAPI. Technical replicates were included in each assay. Nuclei were counted within 25 fields of view in each insert (5 × 5 tiles; tile area = 456 µm × 456 µm).

Invasion assays were performed based on Scott *et al* (2011) with some modifications. 24-well plates with 8-µm pore-size inserts were used. Matrigel (BD Biosciences, cat# 354324) was used at 50% dilution with PBS. 100 µl of matrix solution was added to the inserts and let to solidify. 40,000 cells (in 100 µl regular media) were plated on the upward facing underside and allowed to attach for 4 h. Subsequently, transwells were inverted, the underside was washed with serum-free media, and transwells were transferred in wells containing 500 µl of serum-free media. 100 µl of 10% FBS-containing media were added on top of the solidified Matrigel, and cells were allowed to invade for 48 h. Transwells were fixed in 4% PFA/0.2% Triton X-100 in PBS for 1 hr at room temperature, and nuclei were stained with DAPI overnight.

## Proximity ligation assay

Cells plated on collagen IV (Sigma, Cat# C5533)-coated coverslips were fixed with 4% formaldehyde in PBS for 15 min and then permeabilized with 0.5% Triton X-100 in PBS for 5 min at room temperature. Cells were subsequently blocked in Duolink blocking solution (Sigma-Aldrich, cat# DUO92101) for 1 h at 37°C and incubated with a pair of primary antibodies diluted in Duolink antibody diluent (in a humidified chamber for 1.5 h at room temperature). Antibodies used were anti-Rab13 (1:200; Novus Biologicals, cat# NBP1-85799), anti-GFP (1:8,000; Invitrogen, cat# A11122), and anti-RABIF (1:1,500; Santa Cruz Biotechnology, cat# sc-390759). After washing, PLA probes were applied in 1:10 dilution using the diluent buffer provided in the Duolink *In Situ* Red Kit (Sigma-Aldrich, cat# DUO92101). Incubations and subsequent ligation and amplification steps were performed according to the manufacturer's instructions. After the final washes, cells were post-fixed for 10 min at room temperature with 4% formaldehyde in PBS, stained with Alexa Fluor-488 phalloidin (Thermo Fisher, cat# A12379) in blocking buffer for 30 min, and mounted using Duolink PLA Mounting Medium with DAPI. For quantitation, given the variable degree of

amplification that can result from PLA, we avoid measuring intensities and instead count the number of individual PLA dots per cell. Any dot whose intensity is > 3 SDs above the background noise is considered a positive PLA dot.

### Co-translational association of RABIF with RAB13 RNA

Cells treated with vehicle or 50 μg/ml puromycin for 15 min (to disrupt sites of translation) were lysed as described above and processed for RABIF immunoprecipitation using Protein G Dynabeads (Fisher Scientific). After multiple washes, immune complexes were eluted in the same buffer supplemented with 1% SDS by rocking at RT for 15 min. Total RNA present in the eluate or the extract (input) was isolated using TRIzol LS (Thermo Fisher Scientific) and reverse-transcribed using the iScript cDNA Synthesis Kit (Bio-Rad). Fixed amount of spike RNA (β-globin) was added in each sample to correct for potential losses during these steps. The abundance of GFP, GAPDH, and β-globin RNA was then quantified using Digital Droplet PCR (Bio-Rad). Values in each eluate were normalized to the corresponding values in the input, and the enrichment of GFP-RAB13 RNA over GAPDH was assessed.

### Imaging and image analysis

Images of all fixed samples were obtained using a Leica SP8 confocal microscope, equipped with a HC PL APO 63× oil CS2 objective. Z-stacks through the cell volume were obtained, and maximum intensity projections were used for subsequent analysis. Image analysis was performed using ImageJ or RDI Calculator (Stueland *et al*, 2019).

For live imaging of Lifeact-GFP-expressing cells, cells were plated on collagen IV-coated coverglass and were imaged on a Leica SP8 confocal microscope equipped with HC PL APO 63× oil CS2 objective, at constant 37°C temperature and atmospheric air. The 488 nm laser line was used for illumination, and z-stacks through the cell volume were acquired every 1.5 min over a period of 1 h. Maximal intensity projections were produced, and edge velocity was determined using the ADAPT ImageJ plug-in (Barry *et al*, 2015) by thresholding the image and using the plug-in's default settings. Protrusion and retraction velocities were averaged for each cell.

For live imaging of cherry-NLS-expressing cells, cells were plated on collagen IV-coated coverglass and were imaged on an Olympus IX81 microscope equipped with a 10× Phase U Plan Fluorite Dry Objective, a pE-300ultra Illumination System, an ORCA-Flash4.0 v3 sCMOS Camera, and an Okolab cage incubator enclosure. Time-lapse images were taken every 5 min over a period of 10 h. Nuclei were tracked using the built-in Manual Tracking ImageJ plug-in. The resulting XY coordinates for each cell track were imported into a DiPer macro (Gorelik & Gautreau, 2014) to calculate average cell speed.

### Statistical analysis

The normality of distributions was assessed using the D'Agostino and Pearson test or Shapiro–Wilk test (GraphPad Prism). Normally distributed datasets were analyzed using parametric statistical tests. Datasets deviating from a normal distribution were analyzed using non-parametric tests. The statistical test used in each case, as well as the sample sizes and number of replicates, is indicated in the respective figure legends.

## Data availability

The mass spectrometry-based proteomics data have been deposited in the ProteomeXchange Consortium via the MassIVE partner repository and are available via ProteomeXchange with identifier PXD020682 (http://proteomecentral.proteomexchange.org/cgi/GetDataset?ID=PXD020682).

**Expanded View** for this article is available online.

## Acknowledgements
We would like to thank the CCR Genomics Core at National Cancer Institute/ NIH for help with ddPCR and the NCI LGI Flow Cytometry Core for help with cell sorting and flow cytometry. This work was supported by the Intramural Research Program of the Center for Cancer Research, NCI, National Institutes of Health (S.M.; 1ZIA BC011501).

## Author contributions
SM and KM conceived the study. SM, KM, and MLH contributed to methodology. KM, MS, ANG, TW, LMJ, and SM performed the formal analysis. KM, MS, ANG, TW, and SM participated in investigation. KM and SM wrote the original draft of the manuscript. KM, ANG, TW, MLH, and SM reviewed, and edited the manuscript.

## Conflict of interest
The authors declare that they have no conflict of interest.

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
