## [Review Process File · The EMBO Journal]

RNA localization and co-translational interactions control RAB13 GTPase function and cell migration

Konstadinos Moissoglou, Michael Stueland, Alexander Gasparski, Tianhong Wang, Lisa Jenkins, Michelle Hastings, and Stavroula Mili

DOI: [10.15252/emboj.2020104958](https://doi.org/10.15252/emboj.2020104958)

Corresponding author(s): Stavroula Mili (voula.mili@nih.gov)

Review Timeline:

Submission Date:	10th Mar 20
Editorial Decision:	20th Apr 20
Revision Received:	8th Jul 20
Editorial Decision:	30th Jul 20
Revision Received:	7th Aug 20
Accepted:	14th Aug 20

Editor: Elisabetta Argenzio

Transaction Report:

Thank you for submitting your manuscript entitled "Local RNA translation controls cell migration and Rab GTPase function" [EMBOJ-2020-104958] to The EMBO Journal. Your study has been sent to three reviewers for evaluation, whose reports are enclosed below.

As you can see, the referees consider the work potentially interesting. However, while referee #1 is overall positive and requests you to address only minor points, reviewer #2 and #3 raise several criticisms that need to be solved before they can support publication in The EMBO Journal. In particular, referee #2 requests you to investigate Rab13 protein level and cellular localization in the presence of the chemoattractant and co-treating with a protein synthesis inhibitor. Also, this referee asks you to provide insight into the role of Rab13 post-translational modifications in peripheral versus perinuclear localization of Rab13. Referee #3 requires you to provide further insight into i) Rab13 local translation, ii) Rab13 RNA localization not contributing to the steady-state pool of Rab13 protein; and iii) localization important for the phenotype and the partial loss of protein function.

In addition to the points listed above, addressing all major and minor referees' criticisms will be essential for the publication of your work in The EMBO Journal. I should also add that it is our policy to allow only a single round of major revision. Therefore, acceptance of your manuscript will depend on the completeness of your responses in this revised version.

We generally grant three months as standard revision time. As we are aware that many laboratories cannot function at full capacity owing to the COVID-19 pandemic, we may relax this deadline. Also, we have decided to apply our 'scooping protection policy' to the time span required for you to fully revise your manuscript and address the experimental issues highlighted herein. Nevertheless, please inform us as soon as a paper with related content published elsewhere.

Before submitting your revision, primary datasets (and computer code, where appropriate) produced in this study need to be deposited in an appropriate public database (see <http://msb.embojournal.org/authorguide#dataavailability>). Please remember to provide a reviewer password if the datasets are not yet public. Include a "Data availability" section even if there are no primary datasets produced in the study.

I thank you for the opportunity to consider this manuscript and look forward to your revision.

REFEREE REPORTS

Referee #1:

The paper by Moissoglu et al. addresses the long standing problem of the function of mRNA localization within protrusions of mammalian cells. In a comprehensive and well-controlled study, the authors demonstrate that localization of Rab13 mRNA at these sites is important for cell migration and that this is associated with an unexpected role in driving local, co-translational interaction of the G-protein with a specific GEF, RABIF. The authors reach these conclusions by mapping RNA localization elements and interfering with their function, followed by in depth molecular and cellular analysis of the consequences. It has been widely observed that some mRNAs are targeted to sites that do not correspond to where the bulk of the protein product accumulates. To my knowledge,

this is the first time that it has been clearly demonstrated that such a scenario can involve distinct properties being imparted on a locally translated pool of protein. Therefore this is an important study, which is likely to have a large impact. I only have a few minor points that need to be addressed, which should not require new data collection.

1. There is a preprint from the Herbert lab (doi: 10.1101/374850) that deletes the Rab13 mRNA localization signal in zebrafish and shows an effect on cell migration in vivo (without determining the underlying molecular cause). This study, and its complementary conclusions, such as mentioned in the Discussion.
2. The second part of the title is rather general. Readers are more likely to be drawn in if the aspect of co-translational complex assembly can be introduced.
3. In some figures the raw values are shown as individual points (always my preference) and in others they are not. What is the justification for this?
4. The authors state the data in Figure 1D are from two experiments per sample. Statistical evaluations therefore don't seem appropriate (best to show the raw values and allow the reader to judge if the effect is convincing)
5. Figure 6D and E: The y-axis states Rab13 activity but what is being measured is the (normalized) interaction with the RBD MICAL-L1 as a proxy for activity. The y-axis should be labelled directly with what is being measured, possibly with 'Rab13 activity' in parentheses, to assist the reader.
6. Figure 7B: The y-axis is potentially misleading as the scale is different above and below the cropped region. This should be corrected.

Referee #2:

RNAs can be tightly localized at subcellular level, however, the functional role of their specific distributions in different subcellular domains is largely unknown. In this study, Moissoglu and colleagues investigate the cellular distribution of the Rab13 mRNA in the human cell line MDA-MB-231. The authors identify the sequence that specifically mediates the delivery of Rab13 mRNA, but not other ~30 mRNAs, to the periphery versus the perinucleus. They provide evidence that the translation of peripheral Rab13 mRNA does not affect the total Rab13 protein level, or its association to the membrane, but it is required for its GTPase activity and for cell migration (both edge protrusion and retraction), via association with the guanine nucleotide exchange factor RABIF.

This is an excellent study that reports novel and interesting findings. The role of peripherally localized mRNAs in cells is not well understood and a puzzle in the field is that localized mRNAs often encode proteins that are already in abundance and their localized translation contributes little to the global levels of protein. The data presented here indicate that the location of peripheral Rab13 mRNA determines the site of generation of the Rab13-RABIF complex. The study highlights a role for local translation governing a specific protein function with high subcellular localization. The experiments are well executed and controlled, however, the following points should be addressed for accuracy and for a deeper understanding of the molecular mechanism.

Major points

1. Figure 5A and 5C: These experiments are carried out in response to a chemoattractant, whereas the remaining experiments of this study are carried out in basal conditions. This may differentially affect the translational efficiency of Rab13 mRNA. Therefore, for completeness the authors should

investigate the Rab13 protein level and cellular localization in the presence of the chemoattractant and co-treating with a protein synthesis inhibitor, or by puro-PLA, both in control condition and when Rab13 mRNA is mislocalized. A further analysis should also be carried out in edge protrusions versus retractions.

2. Figure 7F: It is interesting that when Rab13 mRNA is mislocalized the association with RAB1F drops, although the protein level is unaltered (Figure 6A). This suggests that the diffusion/transport of perinuclear Rab13 cannot compensate the peripheral Rab13, which might play a specialized function. If peripheral Rab13 is not a specific isoform, a possibility is that Rab13 might potentially undergo post-translational modifications (e.g. Shinde and Maddika, 2018). It would be interesting to screen for Rab13 post-translational modifications in the periphery versus the perinucleus by immunocytochemistry if any antibodies are available, or to pull down the perinuclear versus the peripheral Rab13 mRNA and to perform mass spectrometry in order to investigate the presence of proteins associated to the transcript that may post-translationally modify peripheral Rab13. I recognize that this may be beyond the scope of the current study but it would be valuable if the authors can give some further insight into this question.

Minor points

RAB13 RNA and protein exhibit distinct subcellular distributions:

3. Line 9-12: Firstly, it is unsurprising that Rab13 RNA and protein exhibit distinct subcellular distributions given that several studies have shown in various systems that there is no correlation between RNA abundance and protein amount (e.g. Gygi et al., 1999; Rajasundaram et al., 2014). Secondly, the results are in line with the ones already reported in Figure 7 of the study Moissoglu et al., 2019 indicating that Rab13 translation is less peripheral compared to the Rab13 mRNA signal. Thirdly, a possibility is that the Rab13 mRNAs enriched at the periphery may be translationally up-regulated in response to cue stimulation. In light of these points, the last sentence of the paragraph is too speculative and should be rephrased in a more balanced manner as it seems to assume that the peripheral Rab13 mRNAs are locally translated and the proteins distributed to the perinucleus.

A GA-rich motif within the mouse Rab13 3'UTR is necessary for localization at protrusions:

2. Line 18-21: To be able to make this statement, statistical analyses between the 'Rab13 UTR ($\Delta 1$)' and 'Rab13 UTR ($\Delta 1+2$)' conditions and between each of these conditions and ' β -globin' need to be carried out.

Peripheral RAB13 RNA localization is important for cell migration:

3. Figure 5D: Zoomed images of edge protrusions and retractions should be provided. In addition to measuring the speed, a further analysis of this data should be carried out on the number of protrusions added versus retracted.

4. Figure 5D and 5F: It is interesting that Rab13 mRNA mislocalization or knockdown affects both edge protrusion and retraction, although Moissoglu et al., 2019 reported that Rab13 mRNA translation is silenced at retracted protrusions. This controversial point should be discussed. A possibility is that a specific stoichiometry of Rab13 may underlie the edge protrusion versus retraction, as suggested by the ~20% Rab13 puro-PLA signal detected in retracting protrusions (Figure 8F of Moissoglu et al., 2019).

Mislocalization of RAB13 RNA from the periphery phenocopies acute RAB13 protein loss:

5. Figure 5E-F: A further analysis of this data should be carried out on the number of protrusions added versus retracted.

Discussion

6. Line 39-43: It should be stressed that the experiments reported in this study have been carried out in basal conditions. The amount of protein locally translated could be differentially regulated in response to an extracellular cue in the order of minutes (e.g. Cagnetta et al., 2018). Therefore, this paragraph should be potentially revised in light of the results of the experiment requested above in response to the chemoattractant.

Referee #3:

In the current study, Moissoglu and coworkers investigate RNA localization to protrusions of migrating mesenchymal cells. In particular, the authors focus on RAB13 RNA encoding for a GTPase involved in vesicle-mediated vesicle trafficking. Interestingly, they find the RNA to be enriched at peripheral protrusions, while the expressed protein is mainly found at the perinuclear region. Further, Moissoglu et al. perform antisense experiments against the RNA to specifically interfere with the RNA localization and investigate cell migration. Further, they report that the peripheral RAB13 RNA promotes the local association of RAB13 with the exchange factor RABIF and that this association is co-translational.

Overall, the figures are generally well designed, understandable and provide proper insight into the underlying data. The text is well written and the data presented in a comprehensible manner. However, as the short summary above indicates the individual findings are not (yet) connected, the interpretations of the experiments are not sufficiently backed up by the provided data and the role of translation is currently overemphasized (e.g. see their title). In the remainder, there are detailed comments for the authors to strengthen the focus of the current study and provide additional experiments on translation to back up their claim.

A core claim of this paper (see title) is the effect of local Rab13 translation in the cell periphery on Rab13 protein function. The data strongly indicate and support that Rab13 mRNA is localized. However, convincing data on local translation is missing, but essential for the major claim of the paper (see title). It would be useful, if the authors showed that this RNA actually undergoes local translation. As the authors already have the proximity ligation assay established, a puromycylation-PLA would provide experimental support for their claim. Moreover, the authors have to provide experimental evidence that the localized Rab13 mRNA is indeed translated into Rab13 protein. Localization of a transcript can also provide a way to alter translation dynamics. Are there uORFs that are activated locally? Do ribosomes undergo frameshifting when translating Rab13 mRNA? Since local Rab13 translation does not contribute to the Rab13 protein pool, there might be an alternative role of this particular mRNA.

Second, the authors claim in Fig. 6 that Rab13 RNA localization does not contribute to the steady-state pool of Rab13 protein. This point needs to be addressed in more detail. How stable is Rab13 protein? Could it be that Rab13 mRNA is barely translated in the periphery? Moreover, the authors should provide absolute numbers for perinuclear mRNA dots and in the periphery to get a feeling for the amount of mRNA that is localized. In addition, how sure are the authors that Rab13 mRNA has no function in posttranscriptional gene regulation? Can the authors perform a rescue experiment using a start codon deleted or frameshifted Rab13 transcript?

Third, there is redundancy in Figures 2-4, in which the localization of mmu and hs RAB13 mRNA is

shown to be dependent on a GA motif. This makes reading the paper less thrilling. The authors provide a number of experiments to control for the effects of interfering with the RGAAGRR motif of RAB13. Nonetheless, it would be useful to discuss if this interference were result in indirect effects other than localization in the authors' opinion, e.g. effect on general translation of Rab13. Currently, the authors cannot really claim that the localization is important for the phenotype and the partial loss of protein function.

Fourth, the authors show in Fig. 7 that the interaction between RABIF and Rab13 is impaired in the periphery but not in the cell body upon Rab13 mRNA localization. It might be that Rab13 mRNA is needed for the assembly of RABIF and Rab13. Can the authors comment on this?

Minor comments:

- It would be useful for the reader to see magnified insets of specific regions in microscopical images to better asses the localization of RNA and protein.
- The authors should clearly state for all error bars what is shown (presumably standard error).
- For Fig.1D, the authors should specify why the cells are GFP positive.
- For Fig. 2, it would be interesting to see the effect of $\alpha 2$ alone.
- Fig. 3A is partially redundant with Fig. 2A. You could simplify Fig. 3A by excluding the GA% content diagram.
- For Fig. 3 CD, it would be helpful to add Rab13-PMO#2 directly in the labelling of the figure.
- On page 6 (in relation to Fig 3C), the authors should state clearly in the text and in the figure legend that Cyb5r3 contains RGAAGRR motifs. Similarly, in regards to Fig. 3D and Fig. 4AB, the authors should indicate which RNAs contain GA-rich motifs.
- For Fig. 4C, it would be informative to see a representative example of a western blot.
- For the barplot in Fig. 5C, the authors should include the unit on the y-axis. Presumably arbitrary units (AU).
- For Fig. 4D, the authors should state if the images are an example of a control or RAB13 PMO cell. Moreover, it would be interesting to include image examples of both, especially to compare the edge velocity panel.
- For Fig. 4E, the authors may consider to create a separate panel for the western blot, as the barplot is not a quantification of this blot.
- On page 8, the authors indicate that cells were tracked to assess both speed and directionality. However, no data on directionality is given for this experiment.
- On page 8, for the last sentence the authors could consider limiting their assessment to the observed effects on the actin network.
- For Fig. 7A, it would help to rapidly understand the figure to include a short description of RAB13 T22N in the figure legend.
- For Fig. 7E, it is unclear what the arrows in the control-PMO conditions are pointing to. Perhaps more representative images can be shown. Otherwise the authors should specify better how 'periphery' is defined.
- The authors should include a few sentences in the methods to describe how the proximity ligation assays were quantified.
- In Fig. 8, the authors show association of GFP-Rab13 mRNA with RABIF using RABIF IP. It would be interesting, how the GFP-Rab13 construct including the 3'-UTR and α PMO would be associated with RABIF.

We would like to sincerely thank all Reviewers for their useful and constructive comments. Addressing them has resulted in a clearer and improved manuscript. We list below our specific responses. We hope that you will find the revised version of this work suitable for publication.

Referee #1:

The paper by Moissoglu et al. addresses the long standing problem of the function of mRNA localization within protrusions of mammalian cells. In a comprehensive and well-controlled study, the authors demonstrate that localization of Rab13 mRNA at these sites is important for cell migration and that this is associated with an unexpected role in driving local, co-translational interaction of the G-protein with a specific GEF, RABIF. The authors reach these conclusions by mapping RNA localization elements and interfering with their function, followed by in depth molecular and cellular analysis of the consequences. It has been widely observed that some mRNAs are targeted to sites that do not correspond to where the bulk of the protein product accumulates. To my knowledge, this is the first time that it has been clearly demonstrated that such a scenario can involve distinct properties being imparted on a locally translated pool of protein. Therefore this is an important study, which is likely to have a large impact. I only have a few minor points that need to be addressed, which should not require new data collection.

1. There is a preprint from the Herbert lab (doi: 10.1101/374850) that deletes the Rab13 mRNA localization signal in zebrafish and shows an effect on cell migration in vivo (without determining the underlying molecular cause). This study, and its complementary conclusions, such as mentioned in the Discussion.

We have added a reference to this work in the Discussion (page 15; lines 434-437).

2. The second part of the title is rather general. Readers are more likely to be drawn in if the aspect of co-translational complex assembly can be introduced.

Thanks for the suggestion. We have altered the title to “RNA localization controls cell migration and GTPase function through co-translational interactions”

3. In some figures the raw values are shown as individual points (always my preference) and in others they are not. What is the justification for this?

Generally, we don't present the individual values if the n value is too large or if there are many samples in a graph, because we think in these cases presenting all individual data points becomes overwhelming and is not visually helpful. We had omitted, though, the individual points

in some graphs with lower n values. We have corrected that in Figures 1D, 4C, 5C, 6D, 6E, 7B, and 7C.

4. The authors state the data in Figure 1D are from two experiments per sample. Statistical evaluations therefore don't seem appropriate (best to show the raw values and allow the reader to judge if the effect is convincing)

We have added the individual raw data points in the graph.

5. Figure 6D and E: The y-axis states Rab13 activity but what is being measured is the (normalized) interaction with the RBD MICAL-L1 as a proxy for activity. The y-axis should be labelled directly with what is being measured, possibly with 'Rab13 activity' in parentheses, to assist the reader.

We have modified the graph labels.

6. Figure 7B: The y-axis is potentially misleading as the scale is different above and below the cropped region. This should be corrected.

We have changed the scale and added the individual data points.

Referee #2:

RNAs can be tightly localized at subcellular level, however, the functional role of their specific distributions in different subcellular domains is largely unknown. In this study, Moissoglu and colleagues investigate the cellular distribution of the Rab13 mRNA in the human cell line MDA-MB-231. The authors identify the sequence that specifically mediates the delivery of Rab13 mRNA, but not other ~30 mRNAs, to the periphery versus the perinucleus. They provide evidence that the translation of peripheral Rab13 mRNA does not affect the total Rab13 protein level, or its association to the membrane, but it is required for its GTPase activity and for cell migration (both edge protrusion and retraction), via association with the guanine nucleotide exchange factor RABIF.

This is an excellent study that reports novel and interesting findings. The role of peripherally localized mRNAs in cells is not well understood and a puzzle in the field is that localized mRNAs often encode proteins that are already in abundance and their localized translation contributes little to the global levels of protein. The data presented here indicate that the location of peripheral Rab13 mRNA determines the site of generation of the Rab13-RABIF complex. The study highlights a role for local translation governing a specific protein function

with high subcellular localization. The experiments are well executed and controlled, however, the following points should be addressed for accuracy and for a deeper understanding of the molecular mechanism.

Major points

1. Figure 5A and 5C: These experiments are carried out in response to a chemoattractant, whereas the remaining experiments of this study are carried out in basal conditions. This may differentially affect the translational efficiency of Rab13 mRNA. Therefore, for completeness the authors should investigate the Rab13 protein level and cellular localization in the presence of the chemoattractant and co-treating with a protein synthesis inhibitor, or by puro-PLA, both in control condition and when Rab13 mRNA is mislocalized. A further analysis should also be carried out in edge protrusions versus retractions.

This is a good point. We have performed the suggested experiments. We do not observe any increase in total RAB13 protein levels (through Western blot analysis), or in the amount of RAB13 protein at protrusions (detected by immunofluorescence) upon serum stimulation (5, 20 or 50 min after addition of serum in serum-starved cells). If anything, serum stimulation leads to a decrease in the amount of RAB13 at protrusions, maybe reflecting increased endocytosis upon stimulation. The results are the same when the *RAB13* RNA is mislocalized. Therefore, at least at the level of resolution we can achieve, we do not have any evidence that acute stimulation promotes peripheral RNA translation leading to increased RAB13 presence at protrusions, or that changing the location of the *RAB13* RNA changes its global or local translational efficiency. These data are now presented in Figure S1 and discussed in pages 5 (lines 87-92) and 8 (lines 195-197).

2. Figure 7F: It is interesting that when Rab13 mRNA is mislocalized the association with RABIF drops, although the protein level is unaltered (Figure 6A). This suggests that the diffusion/transport of perinuclear Rab13 cannot compensate the peripheral Rab13, which might play a specialized function. If peripheral Rab13 is not a specific isoform, a possibility is that Rab13 might potentially undergo post-translational modifications (e.g. Shinde and Maddika, 2018). It would be interesting to screen for Rab13 post-translational modifications in the periphery versus the perinucleus by immunocytochemistry if any antibodies are available, or to pull down the perinuclear versus the peripheral Rab13 mRNA and to perform mass spectrometry in order to investigate the presence of proteins associated to the transcript that may post-translationally modify peripheral Rab13. I recognize that this may be beyond the scope of the current study but it would be valuable if the authors can give some further insight into this question.

Indeed, we also think that post-translational modifications (PTMs) might be part of the underlying mechanism and we have spent significant amount of effort exploring this possibility. Specifically, since RAB13 PTMs are not well-understood, we performed mass-spec analysis to identify in an unbiased manner RAB13 PTMs that might be affected by the location of the *RAB13* RNA (i.e. we compared PTMs of GFP-RAB13 carrying the wt versus Δ PMO UTR). We identified a phosphorylation of Ser155, which appeared to be increased when the *RAB13* RNA is perinuclear (GFP-RAB13/ Δ PMO UTR), indicating that it might correlate with reduced RAB13 activity. This phosphorylation site has not been previously described so there are no antibodies that would allow us to independently validate the mass spec findings. We instead tried to assess the functional importance of this modification in RAB13 activation through mutational analysis. We reasoned that if this phosphorylation plays a role in RAB13 activation then introducing a phospho-mimetic mutation (Ser155 to Glu155) in GFP-RAB13/wt UTR should lead to a decrease in GFP-RAB13 activity, while preventing this phosphorylation by introducing an alanine mutation (Ser155 to Ala155) in GFP-RAB13/ Δ PMO UTR should lead to an increase in GFP-RAB13 activity. However, the results of these experiments were inconsistent with these predictions and did not provide any insights relevant to the mechanism described in this manuscript. We, thus, opted to not describe this work in the text. We have however now included in the Discussion a mention of the potential relevance of PTMs in the activation of peripheral RAB13 (pages 14-15; lines 418-420).

Minor points

RAB13 RNA and protein exhibit distinct subcellular distributions:

1. Line 9-12: Firstly, it is unsurprising that Rab13 RNA and protein exhibit distinct subcellular distributions given that several studies have shown in various systems that there is no correlation between RNA abundance and protein amount (e.g. Gygi et al., 1999; Rajasundaram et al., 2014). Secondly, the results are in line with the ones already reported in Figure 7 of the study Moissoglu et al., 2019 indicating that Rab13 translation is less peripheral compared to the Rab13 mRNA signal. Thirdly, a possibility is that the Rab13 mRNAs enriched at the periphery may be translationally up-regulated in response to cue stimulation. In light of these points, the last sentence of the paragraph is too speculative and should be rephrased in a more balanced manner as it seems to assume that the peripheral Rab13 mRNAs are locally translated and the proteins distributed to the perinucleus.

Respectfully, we do not fully agree with some of the above statements. Firstly, the studies referenced above (Gygi et al, 1999; Rajasundaram et al, 2014) explore the correlation between overall RNA abundance and protein amount, in order to answer the question of whether protein levels can be predicted from RNA levels. We are addressing here a different question of whether the subcellular location of an RNA mirrors the distribution of the protein encoded from it. As detailed in the Introduction of our manuscript, few studies have addressed this question,

and they have provided divergent answers; some showing some correlation (Zapullo et al, 2017; Fazal et al, 2019) while others not (Mardakheh et al, 2015). Additionally, all of these studies reach these conclusions based only on global correlations between transcriptomic and proteomic datasets. We thus believe that this is still an unsettled question and that providing experimental evidence for a specific case is important new information.

With regards to the second point, the reviewer suggests that maybe most of the RAB13 protein is translated internally and thus there might not be a big discrepancy between RNA (which is actually being translated) and protein distribution. We understand that the data referenced (Figure 7 of Moissoglu et al 2019) might contribute to that perception. However, we would like to point out that the data in Figure 7 of Moissoglu et al 2019, exaggerated the difference between RAB13 translation (puro-PLA signal) and RAB13 mRNA (FISH signal). One reason for this is that there is significant perinuclear, non-specific, puro-PLA noise, which we could not remove, and which skews the RAB13 translation into appearing more perinuclear. A second reason is that, in the cells used, several protrusions are retracting and (as Figures 8 and 9 of that manuscript show) RAB13 RNAs are silenced at retracting protrusions, thus likely further contributing to the perception of the RAB13 RNA being more peripheral than the RAB13 translation. It is still though conceivable that the presence of retracting regions might also underlie the difference between RNA and protein distribution reported in Figure 1A-C of the current work. The experiment of Figure 1D was performed precisely to address that possibility by looking at RNA/protein distributions in predominantly extending protrusions induced upon brief LPA stimulation. We find that in extending protrusions the RAB13 RNA is enriched but we still cannot detect any RAB13 protein enrichment. We believe these findings are consistent with the idea that at least some of the protein translated from peripheral RAB13 RNA does not persist at the periphery but assumes a steady state perinuclear distribution. This rationale was not explicitly stated before and we have now modified the text to indicate those points (page 5; lines 75-87).

With regards to the third possibility, as detailed in point 1 above, we have not been able to detect any upregulation of peripheral RAB13 translation in response to serum stimulation. Finally, prompted by a question of Reviewer #3, we have also referenced published data showing that the half-life of RAB13 protein is several hours, which we think is also supportive of the idea that in a moving cells, once synthesized the RAB13 protein does not persist close to the site of synthesis. Altogether, we believe that the conclusion we reach provides the simplest interpretation of all available data. We have, however, also toned it down to allow for the possibility of a different type of regulation (page 5; lines 92-100).

***A GA-rich motif within the mouse Rab13 3'UTR is necessary for localization at protrusions:
2. Line 18-21: To be able to make this statement, statistical analyses between the 'Rab13 UTR***

(Δ1)' and 'Rab13 UTR (Δ1+2)' conditions and between each of these conditions and 'β-globin' need to be carried out.

We have added more statistical comparisons and edited the text to more accurately reflect those (page 6; lines 121-124).

Peripheral RAB13 RNA localization is important for cell migration:

3. Figure 5D: Zoomed images of edge protrusions and retractions should be provided.

In addition to measuring the speed, a further analysis of this data should be carried out on the number of protrusions added versus retracted.

Our description of the analysis was probably not clear. Even though in some instances these cells form characteristic retracting tails, overall, they move through extension/protrusion of large lamellipodial regions and retraction of similarly broad areas. This can be appreciated in the provided movies S2 and S3. For this reason, a specific number of protrusions added or retracted is not possible to assess. The analysis we are doing describes the overall movement of the cell edge by calculating the velocity of each point along the cell boundary between successive movie frames, with positive values indicating extension and negative values indicating retraction. The average speed of extension and retraction for each cell is then computed and used to generate the graphs of Fig. 5D and 5F. We have now clarified this in the text (page 9; lines 224-227).

4. Figure 5D and 5F: It is interesting that Rab13 mRNA mislocalization or knockdown affects both edge protrusion and retraction, although Moissoglu et al., 2019 reported that Rab13 mRNA translation is silenced at retracted protrusions. This controversial point should be discussed. A possibility is that a specific stoichiometry of Rab13 may underlie the edge protrusion versus retraction, as suggested by the ~20% Rab13 puro-PLA signal detected in retracting protrusions (Figure 8F of Moissoglu et al., 2019).

We agree that the effect of *RAB13* RNA mislocalization on retraction dynamics might raise some interesting possibilities. The point raised by the reviewer (i.e. that the specific stoichiometry of *RAB13* translation might underlie protrusion or retraction) is noteworthy. We are also considering the idea that the clustering of *RAB13* RNA might itself play a role in retraction. However, even though the above possibilities are intriguing, we think more probable that the effect on retraction might be an indirect manifestation of feedback mechanisms which coordinate protrusion and retraction during cell movement. We have now mentioned these points in the Discussion (page 15; lines 424-431).

Mislocalization of RAB13 RNA from the periphery phenocopies acute RAB13 protein loss:

5. Figure 5E-F: A further analysis of this data should be carried out on the number of protrusions added versus retracted.

Please see response to point 3 above. In these cells, the specific number of protrusions added or retracted is not possible to assess.

Discussion

6. Line 39-43: It should be stressed that the experiments reported in this study have been carried out in basal conditions. The amount of protein locally translated could be differentially regulated in response to an extracellular cue in the order of minutes (e.g. Cagnetta et al., 2018). Therefore, this paragraph should be potentially revised in light of the results of the experiment requested above in response to the chemoattractant.

Please see response to Major point 1 above. We do not observe any detectable increase in global or local RAB13 protein levels upon acute stimulation with serum.

Referee #3:

In the current study, Moissoglu and coworkers investigate RNA localization to protrusions of migrating mesenchymal cells. In particular, the authors focus on RAB13 RNA encoding for a GTPase involved in vesicle-mediated vesicle trafficking. Interestingly, they find the RNA to be enriched at peripheral protrusions, while the expressed protein is mainly found at the perinuclear region. Further, Moissoglu et al. perform antisense experiments against the RNA to specifically interfere with the RNA localization and investigate cell migration. Further, they report that the peripheral RAB13 RNA promotes the local association of RAB13 with the exchange factor RABIF and that this association is co-translational.

Overall, the figures are generally well designed, understandable and provide proper insight into the underlying data. The text is well written and the data presented in a comprehensible manner. However, as the short summary above indicates the individual findings are not (yet) connected, the interpretations of the experiments are not sufficiently backed up by the provided data and the role of translation is currently overemphasized (e.g. see their title). In the remainder, there are detailed comments for the authors to strengthen the focus of the current study and provide additional experiments on translation to back up their claim.

A core claim of this paper (see title) is the effect of local Rab13 translation in the cell periphery on Rab13 protein function. The data strongly indicate and support that Rab13 mRNA is localized. However, convincing data on local translation is missing, but essential for

the major claim of the paper (see title). It would be useful, if the authors showed that this RNA actually undergoes local translation. As the authors already have the proximity ligation assay established, a puromycylation-PLA would provide experimental support for their claim. Moreover, the authors have to provide experimental evidence that the localized Rab13 mRNA is indeed translated into Rab13 protein. Localization of a transcript can also provide a way to alter translation dynamics. Are there uORFs that are activated locally? Do ribosomes undergo frameshifting when translating Rab13 mRNA? Since local Rab13 translation does not contribute to the Rab13 protein pool, there might be an alternative role of this particular mRNA.

Second, the authors claim in Fig. 6 that Rab13 RNA localization does not contribute to the steady-state pool of Rab13 protein. This point needs to be addressed in more detail. How stable is Rab13 protein? Could it be that Rab13 mRNA is barely translated in the periphery? Moreover, the authors should provide absolute numbers for perinuclear mRNA dots and in the periphery to get a feeling for the amount of mRNA that is localized. In addition, how sure are the authors that Rab13 mRNA has no function in posttranscriptional gene regulation? Can the authors perform a rescue experiment using a start codon deleted or frameshifted Rab13 transcript?

Both the first and second points are overlapping and broadly concerned with issues relevant to the translation of RAB13 RNA, so we will address them together here. Indeed, understanding the translational regulation of the RAB13 RNA is fundamental for the conclusions reached in this work. It might have not come across well in the text that we referred for this information to work that we recently published (Moissoglu et al, 2019). In this work, as the reviewer asks, we extensively investigated the translational regulation of RAB13, and other protrusion-localized RNAs, using biochemical analysis (polysome gradients), single-molecule translation imaging reporters, and puro-PLA of endogenous RAB13, in the same cells as the ones studied here. This work showed that RNAs in the periphery are translated; that the translation rate of individual RNAs is not affected by their location in the cytoplasm (peripheral vs perinuclear); and that silencing of the endogenous RAB13 RNA occurs preferentially at retracting protrusions. Therefore, we believe that the evidence we have thus far (using all currently available experimental tools) strongly points to the fact that the peripheral RAB13 RNA is locally translated in actively extending regions. We have now more clearly referenced this work to support the conclusions reached here (page 5; lines 75-77).

The reviewer also raises the interesting question of whether the peripheral RAB13 RNA is actually being translated into RAB13 protein or whether it might have some translation-independent role or undergo some kind of atypical regulation (such as uORF translation or frameshifting in the coding region). The suggested experiment of performing a rescue with a frameshifted RAB13 transcript is important and we have now included that in the manuscript

(Figures 5G and S5) (described in pages 9-10; lines 243-253). Specifically, we now report that the cell migration defect observed upon RAB13 knockdown can be rescued by re-expression of GFP-RAB13 (from a cDNA expressing GFP fused to mouse RAB13 which is resistant to the siRNA used for knockdown; the construct also contains the full length 3'UTR), but cannot be rescued by the same construct containing only an additional frameshift point mutation (which introduces a termination codon at amino acid #10 of the RAB13 coding sequence). These results thus demonstrate that the production of actual RAB13 protein is necessary and functionally important for the studied cell migration phenotype.

We have no evidence for the existence of additional regulatory mechanisms. The predominant mouse and human RAB13 isoforms contain short 5'UTRs that do not include additional ATG codons. The 5'UTR is also not included in the rescue construct described above, suggesting that it doesn't have an essential role in this mechanism. With regards to ribosome frameshifting, Western blot of endogenous RAB13 detects one major band, and the exogenously expressed GFP-RAB13 constructs result in production of a single protein species (e.g. see Fig 7A).

With regards to the discordance between where the RAB13 RNA is located and where the RAB13 protein ends up at steady-state, we think that the simplest interpretation of our data is that the RAB13 protein translated from peripheral mRNAs does not persist at the periphery but assumes a steady-state perinuclear distribution. Various lines of evidence support this thinking: 1) As shown in Fig 1C, there is ca. 2.5x more RAB13 RNA in the periphery, while the RAB13 protein shows the opposite distribution with ca. 3x more protein around the nucleus. At the same time, our data, from Moissoglu et al 2019, indicate that the RNA is similarly translated in both perinuclear and peripheral regions. (see also response to Minor point 1 of Reviewer #2 for additional discussion) 2) Based on datasets of global protein half-life determination, the half-life of RAB13 protein is several hours (with reported values of 18 hrs, 32 hrs or more). The protein exists in the cytosol, in a pool that is expected to diffuse rapidly, and also associates with intracellular vesicles playing a role in vesicle trafficking. Therefore, we think it is reasonable to assume that most of the lifetime of a RAB13 protein would be spent away from its site of synthesis. 3) Reviewer #2 brought up the possibility that cue stimulation might alter the translational efficiency of RAB13 RNA and thus lead to local protein concentration increases at the periphery. We have now tested this possibility (Figure S1), but do not detect any change in total RAB13 protein levels or in the amount of protein at protrusions upon acute stimulation. These points are now presented in the Results (page 5; lines 80-100)

Third, there is redundancy in Figures 2-4, in which the localization of mmu and hs RAB13 mRNA is shown to be dependent on a GA motif. This makes reading the paper less thrilling. The authors provide a number of experiments to control for the effects of interfering with the RGAAGRR motif of RAB13. Nonetheless, it would be useful to discuss if this interference were result in indirect effects other than localization in the authors' opinion, e.g. effect on

general translation of Rab13. Currently, the authors cannot really claim that the localization is important for the phenotype and the partial loss of protein function.

We think that we have provided significant evidence to support this claim. We have tested for other effects and shown that mislocalization does not alter overall RNA levels (Figures 3D and S4) and that it doesn't affect general translation of RAB13 (Figure 4C); the overall RAB13 protein distribution at steady state or upon stimulation (Figure 6A, B and new Figure S1); or the expression level of exogenous GFP-RAB13 (Figure S6B-D). Together with the demonstration that local peripheral interactions between RAB13 and RABIF depend on the location of the RNA, we think these data reasonably support the claim that the localization of the RNA is important for the observed phenotype.

Fourth, the authors show in Fig. 7 that the interaction between RABIF and Rab13 is impaired in the periphery but not in the cell body upon Rab13 mRNA localization. It might be that Rab13 mRNA is needed for the assembly of RABIF and Rab13. Can the authors comment on this?

We have indeed considered this possibility, i.e. that the RAB13 RNA might act as a scaffold to mediate the interaction between RABIF and RAB13 protein, similar to the model proposed by Christine Mayr et al. This model would predict that RABIF interacts with the RAB13 RNA independent of active translation. What we see however, is that the association between RABIF and RAB13 RNA is abolished upon puromycin treatment (Figure 7B). We thus think that these data are consistent with a co-translational interaction of RABIF with nascent RAB13, which does not involve the RAB13 RNA as an active participant of the complex (mentioned now more explicitly in page 12; lines 344-345).

Minor comments:

- It would be useful for the reader to see magnified insets of specific regions in microscopical images to better assess the localization of RNA and protein.

We have added magnified insets in Figures 1, 3, 4 and 7.

- The authors should clearly state for all error bars what is shown (presumably standard error).

We have added a section on statistics in the methods.

- For Fig.1D, the authors should specify why the cells are GFP positive.

We have used for these experiments a GFP-expressing line. Endogenous proteins and RNAs show variable degrees of protrusion enrichment based on their individual regulation, therefore selecting which to use as internal controls can be difficult and biased. We find that using GFP as an exogenous 'inert' RNA/protein provides more confidence in the conclusions. This information is now included in the Methods.

- For Fig. 2, it would be interesting to see the effect of $\Delta 2$ alone.

Yes, indeed. However, given that the ASO experiments indicated that, apart from these specific motifs, additional GA-rich sequences are important for localization we did not think that, at this stage, this experiment would be critical to do. Understanding the exact contributions of particular sequences within the broader localization element would require in depth analysis (both at a primary sequence and probably at a 3D structural level) which is beyond the scope of this work.

- Fig. 3A is partially redundant with Fig. 2A. You could simplify Fig. 3A by excluding the GA% content diagram.

If there are no space concerns, we would opt to keep it to facilitate comparisons of the data between the two figures.

- For Fig. 3 CD, it would be helpful to add Rab13-PMO#2 directly in the labelling of the figure.

We have edited the figure to indicate that.

- On page 6 (in relation to Fig 3C), the authors should state clearly in the text and in the figure legend that Cyb5r3 contains RGAAGRR motifs. Similarly, in regards to Fig. 3D and Fig. 4AB, the authors should indicate which RNAs contain GA-rich motifs.

We have clarified that in the text.

- For Fig. 4C, it would be informative to see a representative example of a western blot.

Western blot data of such samples are already presented in Figure 5E and 6D. We did not add them in this panel to avoid redundancy.

- For the barplot in Fig. 5C, the authors should include the unit on the y-axis. Presumably arbitrary units (AU).

We have edited the graph.

- For Fig. 4D, the authors should state if the images are an example of a control or RAB13 PMO cell. Moreover, it would be interesting to include image examples of both, especially to compare the edge velocity panel.

We have indicated in the legend that the example is from a control PMO cell. We have also added as a supplemental movie an example of a PMO-treated cell (movie S3).

- For Fig. 5E, the authors may consider to create a separate panel for the western blot, as the barplot is not a quantification of this blot.

We have edited the panel and corresponding legend to better indicate that the graph presents migration speeds and not quantification of the blot. Adding a quantification of the Western blot would partly duplicate the data presented in Figure 4C, so we wanted to avoid that.

- On page 8, the authors indicate that cells were tracked to assess both speed and directionality. However, no data on directionality is given for this experiment.

We do not detect any effect on directionality. We have removed the reference to avoid confusion.

- On page 8, for the last sentence the authors could consider limiting their assessment to the observed effects on the actin network.

Even though the cells are labeled with Lifeact-GFP, this experiment is not measuring actin dynamics but rather the overall speed of movement of the cell boundary, which can be influenced by various cytoskeletal elements. We, thus think, it might not be accurate to conclude that the effect is only mediated through the actin network.

- For Fig. 7A, it would help to rapidly understand the figure to include a short description of RAB13 T22N in the figure legend.

We have provided that information in the legend.

- For Fig. 7E, it is unclear what the arrows in the control-PMO conditions are pointing to. Perhaps more representative images can be shown. Otherwise the authors should specify better how 'periphery' is defined.

We added zoomed in insets to better indicate the signal. Our definition of 'periphery' is presented in Figure S9B and we have explained more in the legend how we define it.

Specifically, phalloidin staining is used to discriminate peripheral regions exhibiting denser cortical F-actin staining from perinuclear/cell body regions which exhibit less intense and more uniform staining.

- The authors should include a few sentences in the methods to describe how the proximity ligation assays were quantified.

We have added details in the methods. Given the variable degree of amplification that can result from PLA, we avoid measuring intensities and instead measure the number of individual PLA dots per cell. Any dot whose intensity is >3 SDs above the background noise is considered a positive PLA dot.

- In Fig. 8, the authors show association of GFP-Rab13 mRNA with RABIF using RABIF IP. It would be interesting, how the GFP-Rab13 construct including the 3'-UTR and Δ PMO would be associated with RABIF.

Indeed, that could be informative given also that we do detect perinuclear interaction between the RAB13 and RABIF proteins (Figure 7E and 7F).

If RABIF interacts with the GFP-RAB13/ Δ PMO RNA, this would indicate that co-translational interaction also occurs at perinuclear locations. This would in turn suggest that the dependence on location for RAB13 activity involves additional spatially restricted events (such as for example post-translational modifications that occur specifically at the periphery; see response to major point 2 of Reviewer #2 for our efforts to identify such modifications).

If RABIF does not interact with the GFP-RAB13/ Δ PMO RNA, this would indicate that co-translational interaction does not occur at perinuclear locations. One reason for that could be again local modifications which in this case might kinetically bias the interaction to occur after translation.

In either case, the result would not alter the conclusions reached, namely the dependence of peripheral RAB13-RABIF interaction on localization of Rab13 RNA, and therefore we have opted to only add a brief reference in the Discussion to the question of whether perinuclear RAB13-RABIF interaction also occurs co-translationally (page 15; lines 420-423).

Thank you for submitting your revised manuscript. Your study has been re-reviewed by the original referees and we have now received their reports, which are enclosed below.

As you will see, referee #1 and #2 find that their criticisms have been sufficiently addressed and recommend the manuscript for publication. However, referee #3 asks you to clearly distinguish your past and present findings on the translational regulation of Rab13 mRNA in the introduction section. In addition, this referee feels that the title does not match the main conclusions of the manuscript and requests you to tone it down.

In addition to solving the remaining points by referee #3, there are a few editorial issues concerning text and figures that I need you to address, before we can officially accept your manuscript.

REFEREE REPORTS

Referee #1:

The authors have addressed all of my minor comments satisfactorily and I now recommend publication.

Referee #2:

The authors have addressed all of my comments well. The findings reported in the manuscript are novel and interesting and make an important contribution to the field.

Referee #3:

The authors made an effort to address some of my concerns. Most importantly, they added the suggested rescue experiment with RNAi resistant Rab13. From this point of view, this clearly strengthens the story. Second, the authors clarified to me that key data on translational regulation of Rba13 mRNA had already been published in their previous paper (Moissoglu et al., 2019). I must have missed this important fact, most likely due to the initial title of the first submission (Local RNA translation controls cell migration and Rab GTPase function). However, when I now checked the revised manuscript, in particular the introduction, I still think that the introduction should be more explicit on this important fact. It is mandatory to outline what has been previously published and what is the experimental rationale of this paper. Only if this distinction has been made, it is possible to build on these key findings and clearly distinguish between past and present findings.

Thirdly, I am not sure that I agree with the new title, as the second part "...through co-translational interactions" now introduces a new claim that is currently not (yet) sufficiently backed up by experiments. Either the authors should significantly tone down this strong claim or they have to add experimental data directly showing co-translational interactions via an RABIF IP using N- and C-terminally tagged Rab13 constructs.

In conclusion, my enthusiasm for the current story is not as strong as it could be since the current experiments do not yet provide direct strong support for the existing claims.

The Authors' have addressed all remaining editorial issues.

I am pleased to inform you that your manuscript has been accepted for publication in The EMBO Journal.

Corresponding Author Name: Stavroula Mili

Manuscript Number: EMBOJ-2020-104958R